

# **Regional sea level budget**
# **over 2004-2022**

7

Marie Bouih[1], Anne Barnoud[1], Chunxue Yang[2], Andrea Storto[2],

Alejandro Blazquez[3] , William Llovel[4], Robin Fraudeau[1] and Anny Cazenave[3]

1. Magellium, 31520 Ramonville St Agne, France

2. Institute of Marine Science, National Research Council of Italy, Rome, Italy

3. Université de Toulouse, LEGOS (CNES/CNRS/IRD/UT3), 31401 Toulouse, Cedex, France

4. Univ Brest, CNRS, Ifremer, IRD, Laboratoire d'Océanographie Physique et Spatiale (LOPS), IUEM, F29280, Plouzané, France

Ocean Science

Revised version, 30 March 2025

Corresponding author : Anny Cazenave

anny.cazenave@univ-tlse3.fr; anny.cazenave@gmail.com

ORCID 0000-0002-2289-1858

# ABSTRACT

Closure of the regional sea level budget is investigated over the 2004-2022 time span by comparing trend patterns from the satellite altimetry-based sea level with the sum of contributions, i.e. the thermosteric, halosteric, manometric and GRD (Gravitational, Rotational, and Deformational fingerprints due to past and ongoing land ice melt) components. The thermosteric and halosteric components are based on Argo data (down to 2000m). For the manometric component, two approaches are considered: one using GRACE/GRACE-Follow On satellite gravimetry data, and the other using ocean reanalyses-based sterodynamic sea level data corrected for local steric effects. For the latter, six different ocean reanalyses are considered, including two reanalyses that do not assimilate satellite altimetry data. The results show significantly high residuals in the North Atlantic for both approaches. In a few other regions, small-scale residuals of smaller amplitude are observed and attributed to the finer resolution of altimetry data compared to the coarser resolution of data sets used for the components. The observed strong residual signal in the North Atlantic points to Argo-based salinity errors in this region. However, it is not excluded that other factors also contribute to the reported non-closure of the budget in this area.

# 1. Introduction

On interannual to decadal time scales, sea level changes in a specific oceanic region arise from several factors. The global mean geocentric sea level rise is primarily driven by ocean warming, land ice melting, and water exchange with continents. Additionally, local and regional effects contribute, including changes in seawater density caused by variations in temperature and salinity (steric effects), as well as the redistribution of ocean water mass through circulation changes (manometric component, Gregory et al., 2019), and variations in atmospheric loading. Furthermore, changes in the solid Earth's gravity, rotation, and deformation (GRD) occur in response to mass redistributions from past and present-day land ice melt and land water storage changes. These GRD factors include two components : the Glacial Isostatic Adjustment (GIA) effect, which stems from the last deglaciation, and GRD fingerprints, which are associated with contemporary land ice melting and, to a lesser extent, changes in land water storage (Gregory et al., 2019).

In terms of global average, the rate of sea level rise is dominated by ocean warming via thermal expansion of seawater, and land ice melting (from glaciers, Greenland and Antarctica ice sheets), in response to global warming (e.g., Cazenave et al., 2018, Nerem et al., 2018, IPCC, 2019, 2021, Cazenave and Moreira, 2022, Horwath et al., 2022, Llovel et al., 2023). The spatial variations of the rate of sea level rise mainly result from steric effects, with the thermosteric contribution being generally dominant (e.g., Stammer et al., 2013, Hamlington et al., 2020), except in the Arctic where the halosteric effect is important (e.g., Carret et al., 2017, Tajouri et al., 2024).

Focusing on trends, many studies have computed the global mean sea level budget over the altimetry era (i.e., since the early 1990s) by comparing the global mean sea level rise with the sum of the thermal and mass components from independent observing systems (e.g., Dieng et al., 2017, Nerem et al., 2018, WCRP, 2018, Horwath et al., 2022, Chen et al., 2018, 2020, Barnoud et al., 2021, 2023, to focus only on the most recent publications). These studies have shown that at least until 2016, the global mean sea level budget is closed within the data uncertainties. In recent years, some discrepancy has been observed between the altimetry-based global mean sea level and the sum of the Argo-based steric and gravimetry-based mass components (e.g., Chen et al., 2020, Barnoud et al., 2021, 2023, Mu et al., 2024), especially when using the Gravity Recovery And Climate Experiment (GRACE) and GRACE Follow-On (GRACE-FO) satellite data to estimate the total mass contribution to sea level change, instead of individual mass contributions (i.e., glaciers, Greenland and Antarctica ice sheets, land

waters and atmosphere water vapor). At regional scale, the closure of the sea level budget
has been less studied so far. A few recent studies have assessed the closure of the sea level
budget at ocean basin-scale, over the altimetry era (e.g., Rietbrock et al., 2016, Frederiske
et al., 2016, 2018, 2020; Hamlington et al., 2020, Royston et al., 2020, Camargo et al., 2023,
Mu et al., 2024). The regional ocean mass budget has also been investigated (Ludwigsen et
al., 2024). Closure of the regional budget is only observed in some regions but not everywhere.
For example, using altimetry, gravimetry and Argo data over 2005-2015, Royston et al. (2020)
concluded that the regional budget cannot be closed in the Indian-South Pacific region.
Similarly, Camargo et al. (2023) also found non-closure of the regional sea level budget in a
number of oceanic areas. Using machine learning techniques, these authors were able to
identify processes not well captured by the observations that are considered to assess closure
of the regional sea level budget.
In the above studies, closure of the regional budget was assessed by averaging the data either
at ocean basin-scale or smaller scale. In the present study, we revisit the regional sea level
budget over the GRACE/Argo era (starting in 2004) at the local scale, without averaging the
data at the basin-scale. After removing the global mean trend of each component, we focus
on the spatial trend patterns, with a resolution of about 300 km, as allowed by the gridded data
sets considered, an approach not applied in the previous studies. This approach avoids
compensation of spurious positive/negative sub-basin trend patterns and allows for more
precise identification of the areas where the sea level budget is not closed.
For this investigation, we use gridded satellite altimetry data for the observed sea level
changes and Argo data to estimate the thermosteric and halosteric sea level changes. For the
manometric component, two types of data are considered: satellite gravimetry data from the
GRACE and GRACE FO missions as well as ocean reanalyses to estimate the redistribution
of water mass in the ocean (following the same approach as in Camargo et al., 2023, i.e.,
estimating sterodynamic sea level changes corrected for local steric effects; see section 3).
The study period covers the period from January 2004 to December 2022 (although some
data sets end in December 2019; section 3).

## 2. Brief overview of the sea level components at regional scale

### 2.1. Steric component

The steric component includes the effects of ocean temperature and salinity changes. Remote surface wind forcing, heat and freshwater fluxes associated with variations in the overlying atmospheric state are the two main forcing mechanisms causing steric changes (Stammer et al., 2013, Roberts et al., 2016). Wind forcing modifies the ocean circulation which further redistributes heat and water masses. It is the dominant mechanism of interannual to decadal steric changes in many regions, particularly in the tropics (e.g., Timmermann et al., 2010; Merrifield and Maltrud, 2011; Piecuch and Ponte, 2014; England et al., 2014). Wind forcing can also play a role in the extra tropics and at high latitudes (Roberts et al., 2016). Buoyancy forcing, i.e. surface air-sea fluxes of heat and freshwater (due to surface warming and cooling of the ocean, and exchange of freshwater with the atmosphere and land through evaporation, precipitation, and runoff) is important in mid to high latitudes, e.g., in the North Atlantic Ocean (Gulf Stream and North Atlantic subpolar gyre) (Roberts et al., 2016).

Over the altimetry era, regional sea level patterns are dominated by steric changes. In most regions, the thermosteric component by far dominates the halosteric one, except in the North Atlantic Ocean (Llovel and Lee, 2015) and in high latitude areas, e.g., in the northeast Pacific, and particularly in the Arctic (e.g., Carret et al., 2016; Ludwigsen et al., 2022, Tajouri et al., 2024). On interannual to multidecadal time scales, the spatial trend patterns in (thermo) steric sea level are still largely influenced by basin-scale internal climate modes of variability, e.g., El Niño-Southern Oscillation (ENSO), Pacific Decadal Oscillation (PDO), Atlantic Multidecadal Oscillation (AMO), North Atlantic Oscillation (NAO) and Indian Ocean Dipole (IOD). Wind stress changes on such time scales are indeed directly related to climate modes (Han et al., 2017). For example, sea level in the tropical Pacific oscillates from west to east with ENSO (with high/low sea level in the eastern/western part during El Niño/La Niña events), in response to wind-forced propagating waves. In the North Atlantic, surface wind and heat flux partly drive interannual to decadal sea level fluctuations and are associated with the NAO (but changes in the Atlantic Meridional Ocean Circulation also contribute) (Han et al., 2017). In the tropical Indian Ocean, interannual to decadal variability in sea level is strongly influenced by ENSO and the IOD (Han et al., 2017, 2019).

## 2.2. Manometric component

The total manometric sea level change has two components: (1) the total water mass added to the ocean (the latter being called barystatic component) due to land ice melt and to the exchange of water with the continents, and (2) the spatial redistribution of water mass by the ocean circulation (Gregory et al., 2019). The barystatic contribution nearly uniformly covers the oceanic domain rapidly (within a few weeks) via a barotropic global adjustment occurring on short time scales (Lorbacher et al., 2012). Because the global mean trend of each component of the regional sea level budget is removed in this study, the barystatic component (i.e., the global mean ocean mass change) disappears. Compared to steric changes, the manometric sea level change due to water mass redistribution (barystatic contribution removed), plays a smaller role on interannual to decadal time scales, but can be sizeable (e.g., Dangendorf et al., 2021, Wang et al., 2022), in particular at high latitudes and over shallow continental shelves (e.g., Forget and Ponte, 2015; Carret et al., 2021).

## 2.3. Atmospheric loading

On seasonal and longer time scales, sea level responds as an inverted barometer to atmospheric loading (Wunsch and Stammer, 1997) i.e. the sea surface height increases (decreases) by 1 cm if the local surface pressure decreases (increases) by approximately 1 mbar. The atmospheric loading component is quite small compared to the thermosteric one, but it is non-negligible at high latitudes (e.g., in the Arctic Ocean where it can reach 0.3 mm/yr equivalent sea level on interannual to decadal time scales, Proshutinski, 2004). Atmospheric loading can be estimated using e.g. surface pressure data from atmospheric reanalyses.

## 2.4. Gravity, Earth Rotation, and solid Earth Deformations (GRD)

The response of the solid Earth to past and present-day water mass exchange between continents and oceans causes global and regional sea level changes. The GIA results from the ice and water mass redistribution of the last deglaciation. Its effect depends on the Earth's mantle viscosity and deglaciation history. The response of the solid Earth to ongoing land ice melt essentially depends on the elasticity of the lithosphere and mantle, as well as on the amount and location of ice mass loss. These mass redistributions induce changes in the gravity, rotation, and visco-elastic deformations of the solid Earth (Mitrovica et al., 2001, Milne et al., 2009, Stammer et al., 2013). These are the so-called GRD (Gravity, Earth Rotation, and solid Earth Deformations) fingerprints (Gregory et al., 2019). In the literature, the GRD contribution is often separated into the one resulting from the GIA (last deglaciation)

and the contemporary GRD effects, the latter referring to mass redistributions due to present-
day land ice melt and land water storage variations. In terms of global average, the GIA effect
on the absolute sea level change is around -0.3 mm/yr (Peltier, 2004; Tamisiea, 2011, Caron
et al., 2018). Its regional signature is mostly uniform, except in formerly glaciated high-latitude
regions. The contemporary GRD fingerprints produce complex regional patterns : sea level
drops near the melting bodies but sea level rises in the far field (e.g., along the northeast coast
of North America). Several studies have theoretically computed the impact of contemporary
GRD changes on relative and absolute sea levels, by solving the sea level equation, either
assuming a priori current ice sheet mass loss (e.g., Mitrovica et al., 2001, Tamisiea, 2011,
Spada, 2017), or using realistic ice mass loss based on observations from the GRACE satellite
gravimetry mission (Adhikari et al., 2019). Note that the sea level fingerprints associated with
the GIA and the contemporary GRDs are usually expressed in terms of linear trends and have
a small amplitude (<0.5 mm/yr except around the ice sheets where the magnitude increases
to ~1 mm/yr), compared to the observed regional sea level and steric sea level trends of
several mm/yr magnitude. However, with the expected increase of land ice melt in the coming
decades, the contribution of the contemporary GRD fingerprints to regional sea level trends
may become increasingly significant.

# 189 3. Data and methods

## 190 3.1. Data

### 191 3.1.1 Altimetry-based total sea level

Total sea level is routinely observed by satellite altimetry. In this study, we use the daily 1/4°
x 1/4° gridded sea level anomaly data version DT2021 from the Copernicus Climate Change
Service (C3S) (https://climate.copernicus.eu). To ensure the long-term stability of this
altimetry-based data, C3S sea level anomalies rely on two simultaneous satellite missions at
any given time: the successive reference missions (TOPEX/Poseidon, Jason-1, Jason-2,
Jason-3, and Sentinel-6 Michael Freilich) plus an auxiliary mission from the global
constellation. The dataset is corrected for TOPEX-A altimeter drift (Ablain et al., 2017), as well
as for the Jason-3 radiometer drift that impacts the wet troposphere correction (Brown et al.,
2023). The dataset covers the period from January 1993 to December 2023. The C3S dataset
is corrected for GIA using the ICE6G-D model (Peltier et al., 2018). The uncertainty in the rate
of the global mean sea level is estimated to 0.3 mm/yr (Ablain et al.,2019, Guérou et al., 2023).
At regional scale, trend uncertainties are larger, on the order of 1 mm/yr especially in coastal
areas (Prandi et al., 2021).

## 3.1.2 Steric sea level

We compute the Argo-based steric sea level data from the Roemmich-Gilson Argo climatology
of the Scripps Institution of Oceanography (SIO) which provides monthly gridded data of
temperature T and salinity S at a 1° x 1° resolution and 58 depth levels until 2000 meters
(Roemmich and Gilson, 2009) (data downloaded in July 2024). The choice for the SIO product
is motivated by the fact that its post-processing corrects for the salinity drift reported in Argo
floats since 2015, which misleads to a spurious increase reported in the global mean salinity
(Wong et al., 2023, Liu et al., 2020, 2024, Ponte et al., 2021). This salinity drift has a significant
impact on the sea level budget closure (Chen et al., 2020, Barnoud et al., 2021). The SIO
processing methodology considers the most up-to-date delayed-mode Argo profiles which
have been meticulously quality-controlled by a scientist (typically within 1-2 years after the
float transmits the data). In addition, the SIO processing adjusts the real-time Argo profiles
(which have passed through automatic quality-control typically within 24 h) to fit the WOCE
(World Ocean Circulation Experiment) global hydrographic climatology. This specific
processing has the benefit of removing the salinity drift in the SIO steric sea level data (Liu et
al., 2024). The dataset covers the period from January 2004 to December 2022, within the 0–
2000 m depth range, and the latitudes between 66°S and 66°N.
In this study, the deep ocean's contribution to steric sea level is not considered due to its small
magnitude (on the order of 0.1 mm/yr) and possibly high uncertainty (e.g., Purkey and
Johnson, 2010). Based on deep Argo profiles, Lele and Purkey (2024) estimated the deep
ocean steric sea level rise (temperature and salinity contribution) being 0.13 ± 0.16 mm/yr in
the south Pacific Ocean over 2014-2023, confirming the small contribution of the deep steric
sea level rise.
The thermosteric, halosteric and total steric sea level changes are computed from the gridded
temperature and salinity data using the Lenapy library (https://github.com/CNES/lenapy) from
the Centre National d'Études Spatiales (CNES), based on the Gibbs seawater oceanography
toolbox of the 2010 Thermodynamic Equation Of Seawater (TEOS-10).

## 3.1.3 Manometric sea level

The manometric sea level change is estimated using two independent methods.
The first approach relies on satellite gravimetry data from the GRACE mission (2002-2017,
Tapley et al., 2019) and GRACE-FO mission (launched in 2018, Landerer et al., 2020), which
enables to estimate changes in the Earth's gravitational field linked to mass redistribution,
including the regional sea level variations due to GRD effects. Two kinds of GRACE solutions
are considered:
-  (1)  An ensemble mean of so-called mass concentration (mascon) solutions (update
from Blazquez et al., 2018). We use the latest GRACE and GRACE-FO Release 6
mascon solutions from the Center for Space Research (CSR; Save et al., 2016), Jet
Propulsion Laboratory (JPL; Watkins et al., 2015), and Goddard Space Flight Center
(GSFC; Loomis et al., 2019). These mascon solutions are corrected for the GIA effect
using the ICE6G-D model (Peltier et al., 2018), as well as for the geocenter motion
using the correction from Sun et al. (2016). The effects of the ocean dynamics and
atmospheric loading are restored using the GAD product derived from the Atmosphere-
Ocean Dealiasing (AOD1B) models (Flechtner et al., 2015, Dobslaw et al., 2017). To
retrieve the ocean mass contribution comparable to the difference between altimetry
and Argo, the effect of the mean atmospheric pressure over the ocean is removed
using the spatial mean of the GAD product at each month (Chen et al., 2019). The
manometric component is estimated as the mean of these three gridded ocean mass
products, which are given in equivalent water height.

-  (2) An ensemble of 60 spherical harmonic (SH) solutions. This ensemble is derived
from the manometric GRACE-based products (DOI: 10.24400/527896/a01-2023.011
version 4.0) and distributed at AVISO+ (https://aviso.altimetry.fr). This product allows
for uncertainty estimates linked to various stages of GRACE and GRACE-FO data
processing (Blazquez et al, 2018). This ensemble of 60 solutions results from the
combination of five processing centres, three C20/C30 (spherical harmonics of degree
2 and 3 of the gravity field potential) estimates,  two GIA models (ICE6G-D from Peltier
et al., 2018 and the model from Caron et al., 2018), and two levels of denoising and
decorrelation kernel filtering. The geocenter motion is corrected with a model based
on the approach developed by Sun et al. (2016) and Swenson et al. (2008). Each
ensemble member is also corrected for the water vapor mass in the atmosphere using
the C0 from GAA (Chen et al, 2019). For each ensemble member, atmospheric loading
over the ocean is restored using the GAD products (Flechtner et al., 2015, Dobslaw et
al., 2017) to correct for the inverse barometer effect, aligning the ocean mass
variations with satellite altimetry data in which atmospheric loading is already
accounted for.

The two sets of GRACE solutions used here cover the period from January 2004 to
December 2022. In case of missing monthly data, a linear interpolation is applied to
account for the data gaps. However, no interpolation is performed for the ~one-year gap
between GRACE and GRACE-FO.

The second approach follows Camargo et al. (2023) method which derives the manometric
sea level change from ocean reanalyses. Ocean models provide the sterodynamic sea level
change, i.e., the sea level change due to changes in ocean density and circulation, with the
inverse barometer correction applied (Gregory et al., 2019; Storto et al., 2024). The
corresponding manometric component is derived by subtracting the local steric effect to the
reanalysis-based sterodynamic sea level, and adding the contemporary GRD fingerprints
(Gregory et al., 2019; Camargo et al., 2023). It is the approach followed here.
We consider six different ocean reanalyses with different characteristics as listed in Table 1.
GLORYS, C-GLORS, ORAS5, and FOAM use the NEMO (Nucleus for European Modelling of
the Ocean) ocean model and assimilate satellite altimetry-based sea level data. The SODA
reanalysis is based on the MOM (Modular Ocean Model) developed by NOAA (National
Oceanographic and Atmospheric Administration, USA) and does not include altimetry data.
All reanalyses have a spatial resolution of 0.25°. In order to assess the regional sea level
budget with another manometric component independent of satellite altimetry data, we also
consider an ensemble reanalysis at lower resolution and without altimetry data assimilation
(called CIGAR; see Table 1).  This variety of reanalyses offers the opportunity to evaluate the
degree of consistency of the manometric signal from reanalysis-based products.

*Table 1: Characteristics of the six ocean reanalyses used in this study to estimate the*
*manometric sea level change patterns independently from GRACE and GRACE-FO data.*

| *Reanalysis* | Ocean model, Spatial Resolution, End date | Data assimilation of altimetry-based sea level data | References |
|---|---|---|---|
| GLORYS (MOI) | NEMO, 0.25°, 2022 | Yes | Garric and Parrent (2017) |
| C-GLORS (CMCC) | NEMO, 0.25°, 2022 | Yes | Storto and Masina (2016) |
| ORAS5 (ECMWF) | NEMO, 0.25°, 2022 | Yes | Zuo et al. (2019) |
| FOAM (UK Metoffice) | NEMO, 0.25°, 2022 | Yes | Blockley et al. (2014) |
| SODA (Version | MOM4, 0.25°, | No | Carton et al. (2018) |

| 3.4.2, U. Maryland) | 2019 | | |
|---|---|---|---|
| CIGAR (CNR-ISMAR) | NEMO, 1°, 2022 | No | Storto and Yang (2024) |


To compare the manometric component derived from ocean reanalyses with the one based
on GRACE and GRACE-FO, we added the contemporary GRD contribution to the reanalysis-
based manometric sea level change. We used the sea level fingerprint data from Adhikari et
al. (2019), which provides monthly contemporary GRD fingerprints at a 0.5° x 0.5° resolution.
Because the Adhikari et al. (2019) data set ends in 2016, we linearly extrapolated the GRD
fingerprints up to 2022 and added the corresponding trends to the ocean reanalyses-based
manometric trends.
## 3.2. Method
Systematic corrections for both atmospheric loading and GIA effects are applied to altimetry-
based and satellite gravimetry datasets, even though different models are used in each
dataset. The MOG2D (Carrere and Lyard, 2003) and inverse barometer model is used for
altimetry data (www.aviso.altimetry.fr), while the GAD product is used for GRACE and
GRACE-FO data (see section 3.1.3). Likewise, the GIA corrections rely on the ICE6-G model
for altimetry (Peltier et al. 2018), while GRACE datasets use either ICE6-G (Peltier et al. 2018)
or Caron et al. (2018) model.
All datasets were spatially interpolated onto a 1° x 1° grid and were averaged on a monthly
basis. For spatial consistency, a common masking technique was applied to all gridded
components. This mask covers latitudes from 66°S to 66°N, excludes inland seas, and omits
coastal regions where the distance from land is less than 300 km.
All datasets span from January 2004 to December 2022, except for the oceanic reanalyses-
based manometric components for which two study periods were considered, depending on
the dataset availability: January 2004-December 2019 and January 2004-December 2022.
Finally, seasonal signals (annual and semi-annual) were removed at each grid mesh of each
data set through a simple least-squares adjustment of 6-month and 12-month sinusoids, and
a 3-month Lanczos filter was applied locally to each dataset to remove high frequency signals.
The global mean trend of each dataset computed over the study period was also removed
before constructing the spatial trend maps.
For each gridded data set, we computed a trend uncertainty map.
For the altimetry data, we used the trend uncertainties provided by Prandi et al. (2021). These
are based on a statistical computation which estimates via a generalized least-squares
approach the total uncertainty of regional sea level trends due to all sources of errors affecting
the altimetry-based sea level measurements (i.e., orbit, range, geophysical corrections and
intermission bias). In this approach, individual variance-covariance matrices describing time
correlated errors are computed for each source of uncertainty. Uncertainties from all sources
are further combined by summing up the variances. Regional sea level trend uncertainties
provided by Prandi et al. (2021) with this method, applied to altimetry-based sea level grids of
2°x2° resolution over 1993-2019 are on the order of 1 mm/yr or less (1-sigma). The largest
errors are located along the continental coastlines.
Since the SIO temperature and salinity data are not provided with uncertainties, we computed
trend uncertainties for the thermosteric and halosteric components by considering the
dispersion between two thermosteric/halosteric products: the SIO product used here and the
EN4 T/S database, version 2.2 (https://www.metoffice.gov.uk/hadobs/en4/download-en4-2-2;
Good et al., 2013). We estimated the thermosteric and halosteric trend uncertainty at each
grid mesh from the dispersion between the two data sets (SIO and EN4) around the mean.
For the total steric, thermosteric and halosteric trend uncertainties were quadratically
combined.
As the ensemble mean of the 60 SH GRACE solutions is provided with trend uncertainties,
(adapted from Blazquez et al., 2018), these are used here for both GRACE-based manometric
components (i.e., the ensemble mean mascon and the SH solutions).
Finally, for the uncertainties of the residual trend map (based on GRACE for the manometric
component), we quadratically combined trend uncertainties of all components at each grid
mesh.
Please note that the reanalyses used here do not provide uncertainty estimates.
In Figure S1 of the Supplementary Information (SI) are shown trend and associated trend
uncertainty maps for the altimetry-based sea level, components and residuals. Note that for
altimetry-based sea level and components, the trend uncertainty map includes the global
mean trend uncertainty (of smaller magnitude than the regional trends). Uncertainties
correspond to 1-sigma errors.

# 4. Results: Regional sea level budget with GRACE and Argo

## 4.1. Trend patterns in observed sea level, components and residuals

Figure 1 shows the maps of altimetry-based sea level trends, of the GRACE- and Argo-based component trends, and of the residual trends (i.e., trend differences between altimetry-based sea level and sum of components). Hatched areas on the figures correspond to regions where the signal to noise ratio is not significant. This is based on comparing at each grid mesh, the observed trend with the trend uncertainty (shown in Figure S1). From Figure 1, we note that except for the elongated negative pattern east of the Philippines in the western tropical Pacific, the altimetry-based sea level trends are significant everywhere. Concerning the GRACE-based manometric component, trends are not significant over a large portion of the northeast Atlantic. The halosteric map displays a few regions where the trends are not significant. Most are located in the southern hemisphere. Accordingly, these translate into the total steric map. Finally, the residual trend map shows that the signal is significant essentially in areas where the residual trends are positive. Over several hatched areas, the residual trends are not significantly different from zero. This concerns most of the Pacific Ocean and a portion of the South Atlantic Ocean.

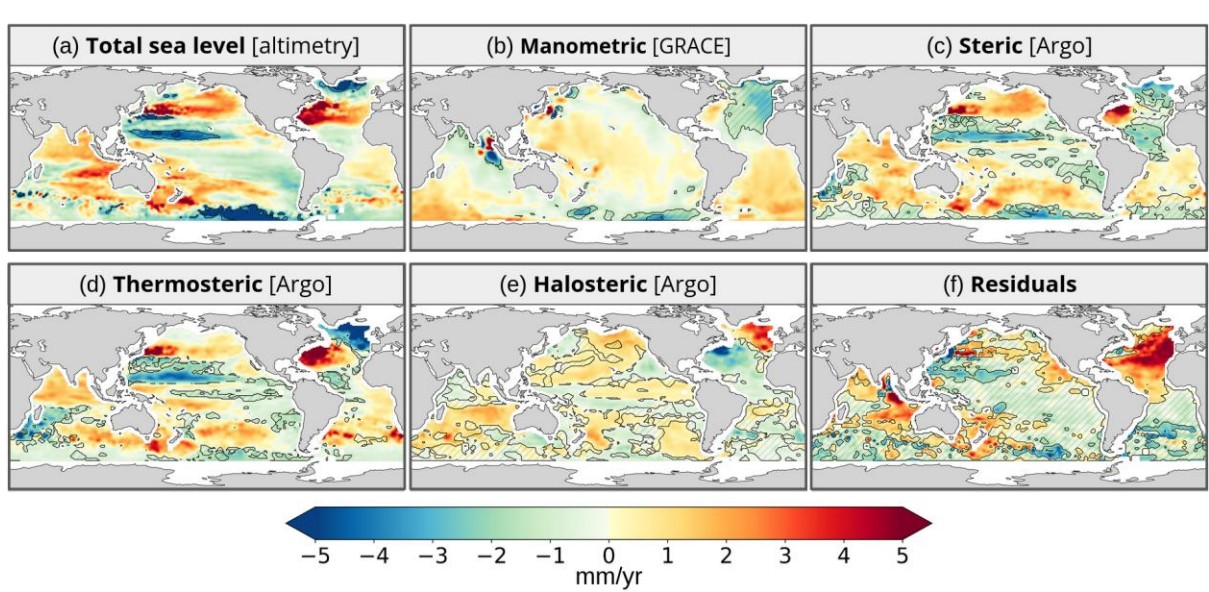

*Figure 1: Sea level trends over January 2004 to December 2022 in total altimetry-based sea*
*level (a), manometric component based on GRACE mascons (b), Argo-based total steric (c),*
*thermosteric and halosteric components (d and e) and budget residual trends (observed sea*
*level minus sum of components) (f). The hatched areas correspond to regions where the signal*
*trend value is not significant compared to the corresponding trend uncertainties.*

Visual inspection of Figure 1 confirms earlier findings, i.e., the observed regional trend patterns
are dominated by the thermosteric trend patterns (as expected; e.g., Stammer et al., 2013;
Hamlington et al., 2020; Cazenave and Moreira, 2022). In the North Atlantic, thermosteric and
halosteric trends have opposite signs. Except for two spots of high signal along the coasts of
north Indonesia and Japan due to the solid Earth response to the Sumatra and Tohoku
earthquakes in 2004 and 2011 respectively (not removed here), the manometric trend map is
dominated by large-scale patterns, positive over almost the whole Pacific, as well as over the
south Atlantic Ocean and southwest Indian Ocean. The absence of small-scale patterns likely
results from the lower resolution of GRACE and Argo data compared to other data sets.
The residual trend map shows that in many regions, the sum of components cancels out the
observed trends. This is the case over most of the Pacific Ocean, part of the South Atlantic
Ocean and southwest Indian Ocean. In these regions, the residuals are not significantly
different from zero, which suggests that the regional sea level budget can be considered as
closed.
In the eastern Indian Ocean, along the coast of North Indonesia, the positive residuals
result from the solid Earth signal due to the 2004 Sumatra earthquake, not removed from the
GRACE-based manometric component. The same is true for the positive residuals east of
Japan and associated with the 2011 Tohoku earthquake. Besides these two regions, it is in
the North Atlantic Ocean that the strongest positive residual trends are observed. In this
region, altimetry-based and thermosteric sea level displays positive trends in the western part
and negative trends south of Greenland while opposite patterns are seen in the halosteric
component. The strong residual signal in the North Atlantic is discussed in detail in section 6.


## 4.2. GRACE data assessment
In this section, we explore the impact of the geocenter and GIA corrections applied to GRACE
data on the residual trend map, considering that these two corrections remain imperfectly
known (Blazquez et al., 2018). For that purpose, we decomposed the sea level budget
components into spherical harmonics and computed the residuals for various configurations
of low degree harmonics (see Figure S2 and Table S1 in the Supplementary Information, SI).
Figure S2 and Table S1 show that degree 1,0 (related to the geocenter motion) and degree
2,1 (related to polar motion and GIA correction) harmonics contribute to the high positive
residuals observed in the North Atlantic Ocean.
GRACE data are classically corrected for the geocenter motion when compared with altimetry
data, in order to moving GRACE observations from the centre of mass to the centre of figure
of the reference system, in which the altimetry-based sea level is supposed to be also
expressed after correcting the satellite orbits for the geocenter motion (Alexandre Couhert,
personal communication).
Using the ensemble of 60 spherical harmonic solutions described in Section 3.1, we
constructed an alternative ensemble of 60 solutions without applying the correction for the
geocenter motion, i.e. keeping the GRACE observations in the center of mass reference
system. Comparing these two ensembles of solutions allows us to assess the influence of the
geocenter correction on the manometric component and, consequently, on the residuals of
the sea level budget. Figure 2 shows the impact of the geocenter correction on the manometric
trends as well as on the associated residual trends. Not correcting for the geocenter motion
reduces the residuals observed in the North Atlantic Ocean but increases the residual trends
elsewhere, with larger residuals in almost all other ocean basins. Thus, even if the Sun et al.
(2016)'s geocenter correction may not be optimal, it minimizes the residual trend   , except
in the northeast Atlantic Ocean. This questions the actual referential of altimetry data and the
consistency of the processing between satellite altimetry and satellite gravimetry data.
Normally, this should be consistent as altimetry-based sea level is supposed to be expressed
in a center of figure reference frame, like the GRACE data after correcting for the geocenter
term. However, the way the geocenter is corrected in the orbits used in the altimetry-based
processing could still be an issue (Alexandre Couhert, personal communication). Besides, if
no geocenter correction is applied, the GRACE-based manometric component   displays
large scale signals not observed by altimetry data, so that the corresponding residual trends
(Figure 2d) also present unrealistic large-scale signals.

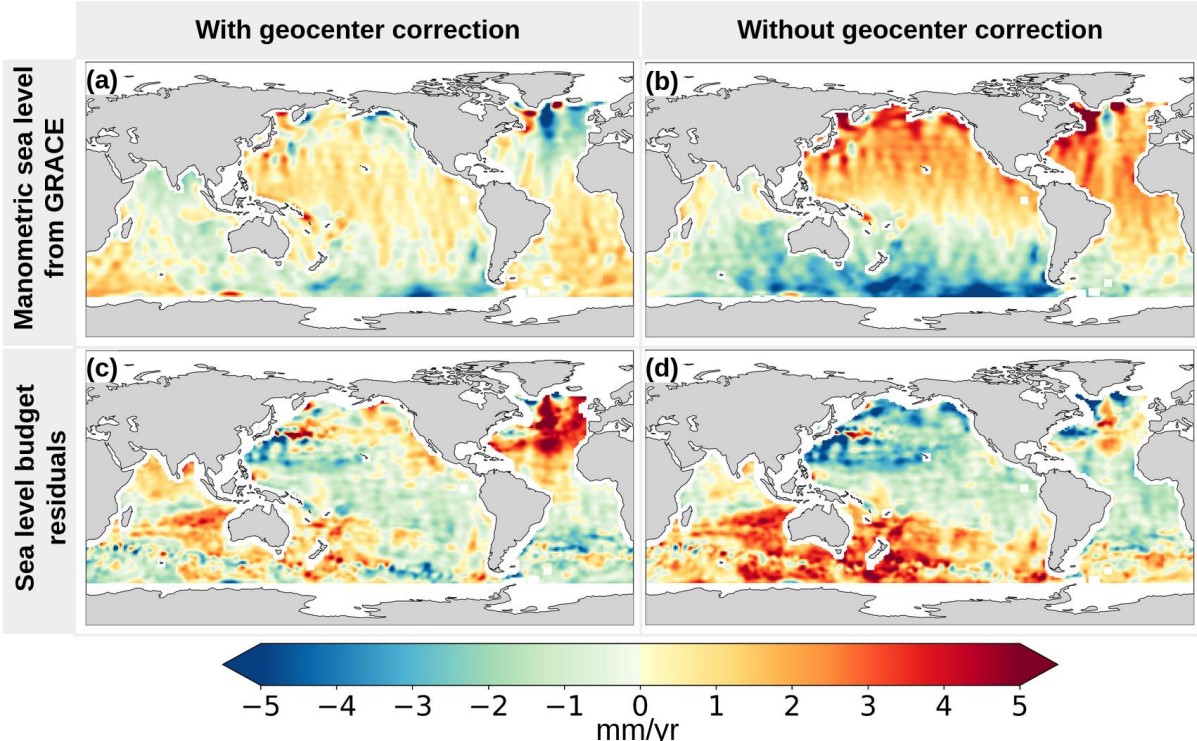

*Figure 2: Sea level trends of GRACE-based manometric component and corresponding budget residuals with and without the geocenter correction. (a) Manometric sea level trend map with the geocenter correction, (b) Manometric sea level trend map without the geocenter correction, (c) Sea level budget residual trend map computed with the manometric component corrected for the geocenter, (d) Sea level budget residual trend map computed with the manometric component not corrected for the geocenter.*

To estimate the impact of the GIA corrections on the manometric component and budget residuals, we further formed two separate subsets of 30 solutions each, for each GIA model (Peltier et al, 2018 and Caron et al., 2018), applying the Sun et al. (2016)'s geocenter correction to each subset. Unlike for the geocenter case, no significant difference was observed.

# 5. Regional sea level budget using ocean reanalyses for the manometric component

As explained in section 2, according to Gregory et al. (2019) and Camargo et al. (2023), the dynamic ocean mass redistribution due to ocean circulation changes can be estimated from the sterodynamic sea level corrected for local steric changes. Thus, in this study, we apply

Camargo et al. (2023)'s approach and use ocean reanalyses to estimate the manometric
component, in order to further assess closure of the regional sea level budget.
As detailed above (section 3), six different ocean reanalyses have been considered over their
common period from January 2004 to December 2019. However, to compute the ensemble
mean reanalysis, we discarded FOAM because of its spurious trends in the South Atlantic and
South Indian Ocean. Figure 3 shows the reanalysis-based manometric trend maps over 2004-
2019, for each of the six data sets, as well as the ensemble mean based on CIGAR, C-
GLORS, GLORYS, ORAS5 and SODA.  The manometric components based on the two sets
of GRACE solutions, restricted to this study period, are also shown.
No uncertainties are provided with the reanalyses data so that it is not possible to highlight the
areas where the signal is significant for the individual cases (Figure 3a-d). This can be done
however for the ensemble mean (Figure 3g) where the errors are estimated from the
dispersion of the five reanalyses (CIGAR, C-GLORS, GLORYS, ORAS5 and SODA) about
the mean.

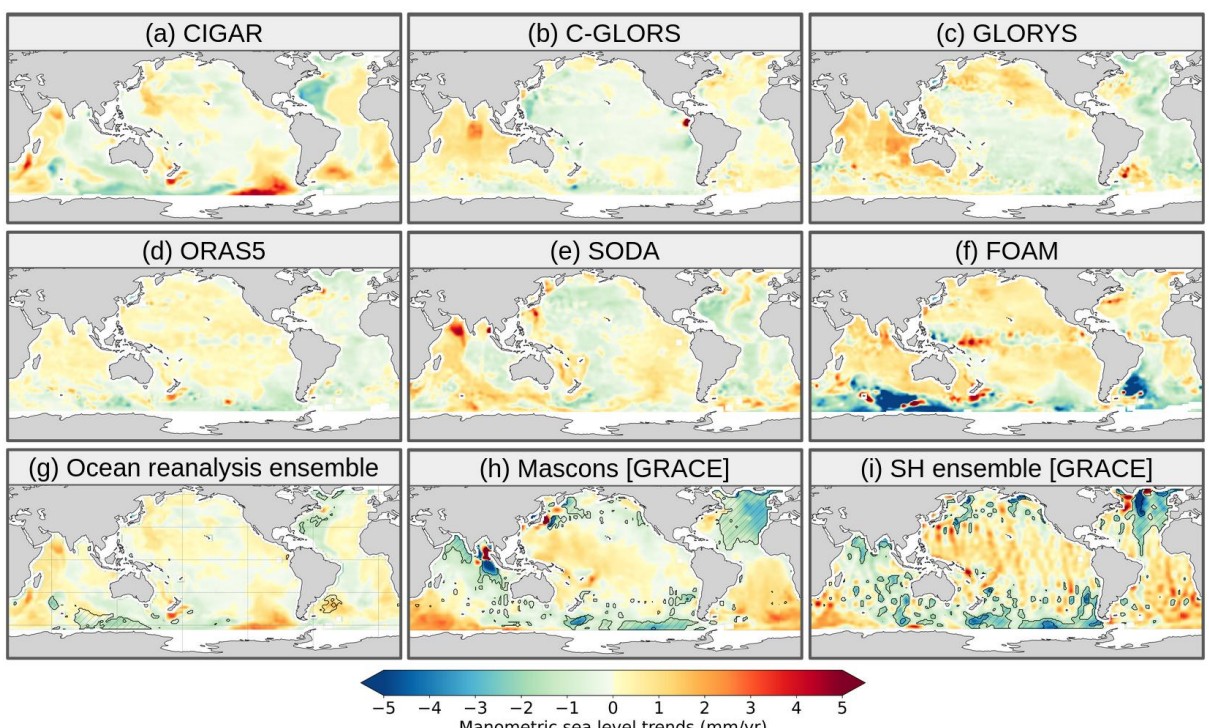


*Figure 3     : Reanalysis-based manometric trend maps over January 2004-December*
*2019 for each of the six ocean reanalyses: CIGAR, C-GLORS, GLORYS, ORAS5,*
*SODA and FOAM (panels a to f). Panel (g) shows the ensemble mean of the five*
*reanalyses (CIGAR, C-GLORS, GLORYS, ORAS5 and SODA). Panels h and i refer*
*to the manometric component from the GRACE mascon and GRACE spherical*
*harmonic (SH) ensembles. Hatched areas in panels 3g, 3h, 3i represent regions where*
*the                signal                is                not                significant.*

The six reanalyses provide quite different manometric trend patterns. The FOAM and
ORAS5 patterns are quite similar in the Pacific and Indian Oceans. In these regions,
C-GLORS, CIGAR and GLORYS show rough agreement. SODA's patterns differ from
the other reanalyses everywhere, although they look similar to CIGAR in the Indian
Ocean. As mentioned above, the FOAM-based manometric map shows spurious high
trends in the South Atlantic and South Indian Ocean. This is why it is not included in
the ensemble mean.
Comparing ensemble mean reanalyses-based manometric map with the GRACE-
based manometric maps, we note that : (1) the spatial patterns of the reanalysis-based
manometric trend map have generally slightly lower amplitude than the GRACE ones
in many regions, except around Antarctica, and (2) the manometric spatial trends of
the ensemble mean reanalyses and GRACE have opposite signs in many regions,
including in the North Atlantic.
The regional budget has been computed with each of the six reanalyses as well as for
the ensemble  mean (FOAM excluded), all other data being kept unchanged. The
corresponding residual trend patterns are shown in Figure 4    .




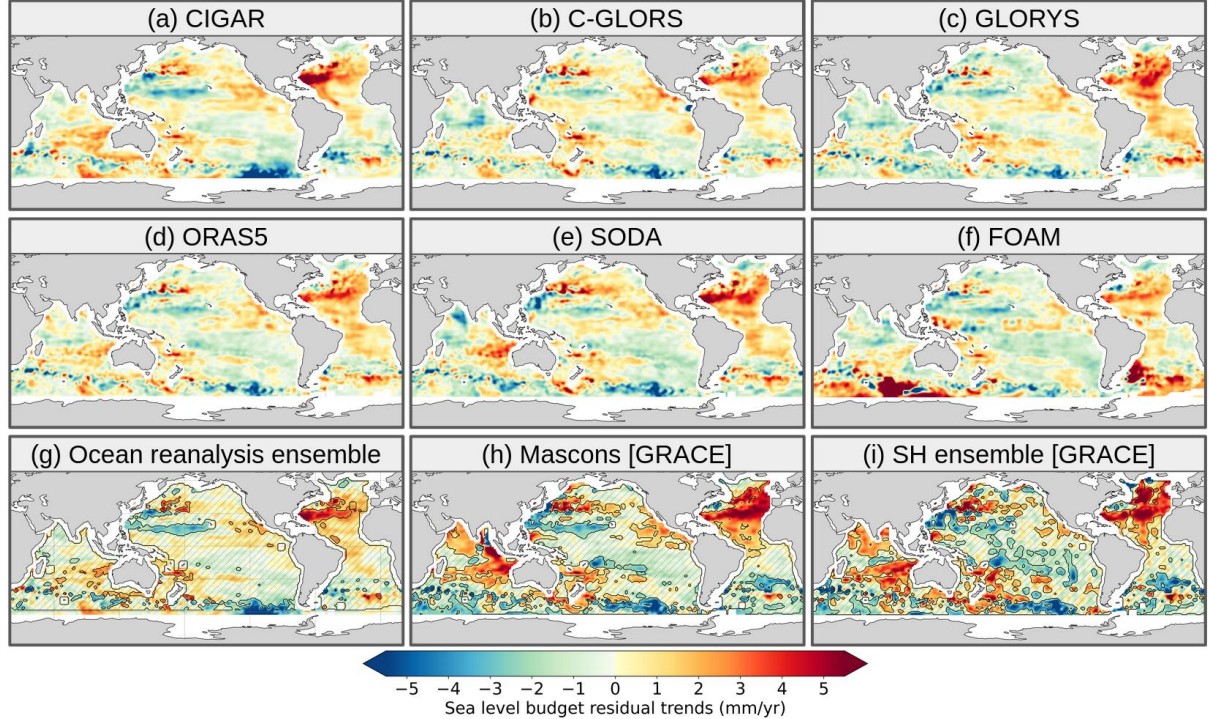


*Figure 4: Residual trends of the regional sea level budget computed with each of the six reanalyses-based manometric components, as well as with the ensemble mean (FOAM excluded) (panels a to g) (with altimetry-based and steric components unchanged). GRACE-based residual trends for both mascon and spherical harmonic solutions are also shown (panels h and i). The period of analysis here is from January 2004 to December 2019. Hatched areas in panels 4g, 4h, 4i represent regions where the signal is not significant.*

The CIGAR, C-GLORS, SODA, ORAS5, and GLORYS reanalyses give very similar
residual trend patterns, even though SODA uses completely different ocean model
and data assimilation schemes, and no altimetry-based sea level data are assimilated.
Note that CIGAR also does not assimilate altimetry-based sea level anomaly data, but
it is forced by the latest atmospheric reanalysis from ECMWF (unlike the other
reanalyses) and embeds a daily varying runoff dataset for freshwater discharge into
the oceans. Again, one outlier is FOAM which shows strong positive residual trends in
the Southern Atlantic and South Indian Ocean. The ensemble mean residual trend
map (FOAM excluded) displays slightly lower signal than the GRACE cases (compare
panel 4g with panels 4h and 4i in Figure 4). What is striking is that the two approaches
(reanalyses and GRACE) show positive residual trends in the North Atlantic. However,
the residuals are significantly stronger using GRACE, especially in the northeastern
part of the Atlantic Ocean. This will be discussed in section 6.
Table 2 shows the root mean square (RMS) of the gridded residuals trends over 2004-
2019, averaged over the Pacific, Indian, North and South Atlantic Oceans for the
reanalyses and GRACE cases.
If we exclude FOAM which displays higher RMS in the Indian and South Atlantic
oceans than other reanalyses, Table 2 clearly shows systematically higher RMS (in
the range 2-3 mm/yr) in the North Atlantic Ocean for the C-GLORS, SODA, GLORYS
and ORAS5 reanalyses as well as for GRACE mascons.
Table 2: RMS of gridded residuals trends over 2004-2019, averaged over the Pacific,
Indian, North and South Atlantic oceans for the six    reanalyses and GRACE mascons
cases.

| RMS (mm/yr) | Pacific Ocean | Indian Ocean | North Atlantic Ocean | South Atlantic Ocean |
|---|---|---|---|---|
| C-GLORS | 1.48 | 1.46 | 2.02 | 1.35 |
| FOAM | 1.66 | 2.81 | 1.99 | 2.54 |
| SODA | 1.58 | 1.79 | 2.68 | 1.56 |
| GLORYS | 1.34 | 1.61 | 2.49 | 1.72 |
|  |  |  |  |  |
| ORAS5 | 1.39 | 1.38 | 2.15 | 1.37 |
| CIGAR | 1.51 | 1.51 | 2.47 | 1.46 |
| GRACE mascons | 1.51 | 1.78 | 2.92 | 1.43 |


Because four of the reanalyses used above assimilate altimetry data (i.e., C-GLORS,
GLORYS, ORAS5 and FOAM), our approach may introduce some circularity to the
regional sea level budget assessment. This is the reason for also using reanalyses
without altimetry data assimilation (SODA and CIGAR). Here we focus on CIGAR and
extend the study period to December 2022.    Comparing the manometric components
of two reanalyses with and without altimetry data assimilation (e.g., C-GLORS and
CIGAR, noting however that they differ in terms of resolution, configuration and forcing
; Figure 4) shows some differences locally, in particular in the northwestern Atlantic
and North Indian Oceans. However, the residual trend maps are very similar. In Figure
5 are shown manometric and residual trend maps based on CIGAR, GRACE mascons
and GRACE SH ensemble, extended until 2022. Overall, the patterns are qualitatively
similar to those of the shorter period 2004-2019 (Figures 3 and 4). Thus, adding three
more years does not change the previous conclusion, i.e., that significant residual
trends are observed in the North Atlantic (and around Antarctica as well). But the
residual patterns in the North Atlantic are noticeably different between CIGAR and
GRACE, the maximum signal being located in the western part of the basin for CIGAR
and in the eastern part for GRACE.
One may wonder whether the salinity drift observed in some Argo floats as of 2015
has impacted the CIGAR reanalysis since, unlike for altimetry data, T/S data are
assimilated during the reanalysis integration, thus non-linearly interacting with
dynamical processes. In the reanalysis, the treatment of the salinity drift simply
consisted in rejecting data that Argo had flagged for rejection in the delayed mode.
But this may not fully guarantee that all bad salinity data have been discarded.
However, to compute the reanalysis-based manometric component, the local steric
contribution has been removed. Thus, any effect of the spurious Argo salinity drift can
be considered as minimal.


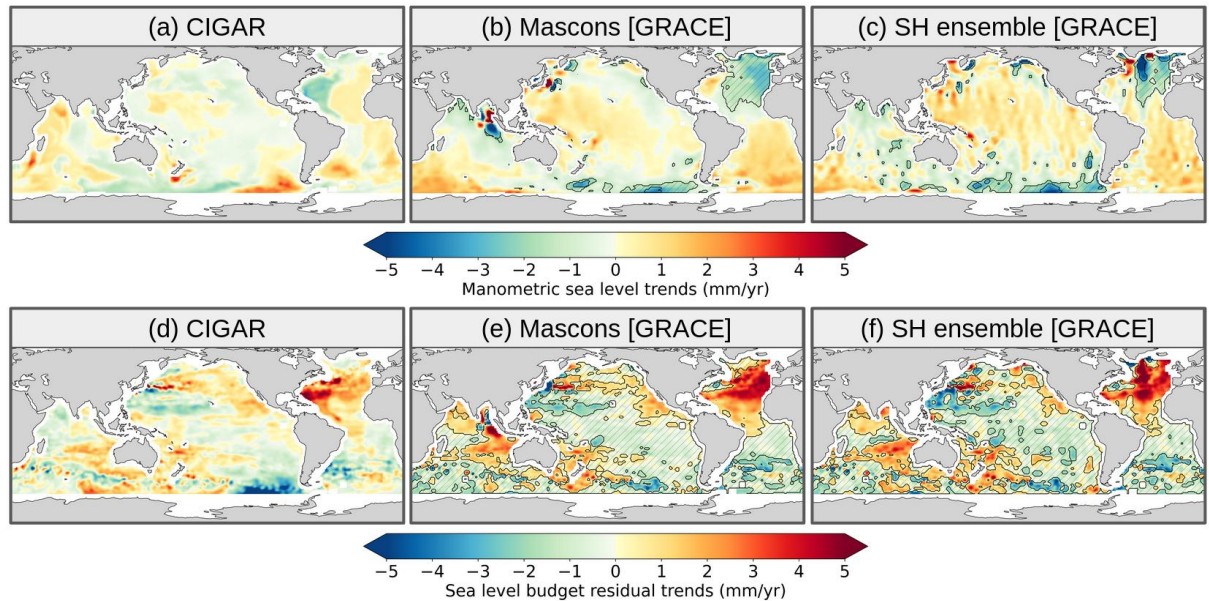

*Figure 5: Manometric component based on the CIGAR reanalysis (reanalysis without altimetry data assimilation) (panel a), and the two GRACE solutions (panels b and c). Sea level budget residuals using the CIGAR-based manometric component (panel d) and the two GRACE manometric components (panels e and f). The period of analysis here is from January 2004 to December 2022. Hatched areas in panels (5b,5c, 5e, 5f represent regions where the signal is not significant.*

# 6. Residual trends in the North Atlantic Ocean

In this section, we focus on the North Atlantic Ocean where significant positive residuals are observed when using either GRACE or the CIGAR reanalysis for estimating the manometric component.

Figure 6 shows each component of the budget over the North Atlantic Ocean over January 2004-December 2022, including the three manometric component estimates : GRACE mascons, ensemble mean GRACE SH and CIGAR reanalysis. Associated residual maps (all components unchanged except the manometric one) are also shown.

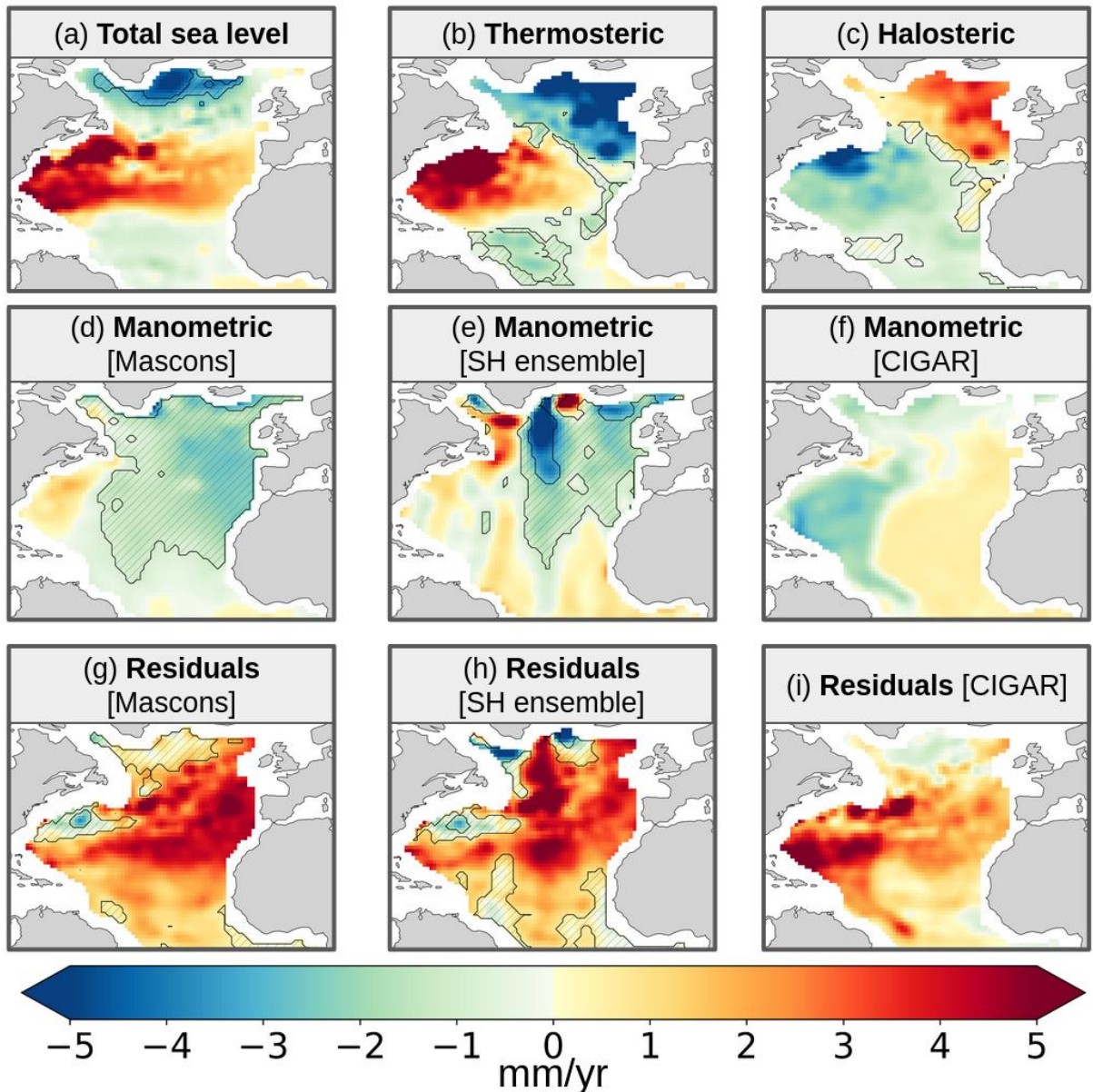

*Figure 6: North Atlantic Ocean sea level trends (mm/yr) over January 2004 to December 2022: Observed altimetry-based sea level (a), Argo-based thermosteric and halosteric components (b,c), manometric components from GRACE mascons (d), ensemble mean GRACE spherical harmonics (e) and derived from CIGAR reanalysis (f). Sea level budget residuals (observed sea level trends minus sum of component trends) using GRACE mascons (g), ensemble mean GRACE spherical harmonics (h) and CIGAR (i). Hatched areas represent regions where the signal is not significant.*

The North Atlantic Ocean residuals of all three manometric component cases (GRACE mascons, ensemble mean GRACE spherical harmonics and CIGAR) show a positive signal. However, the patterns are significantly different between the reanalysis and GRACE cases. They are localized in the western part of the tropical North Atlantic Ocean with the CIGAR

reanalysis and in the eastern part (between the Gibraltar Strait and the Gascogne Gulf)    with
GRACE mascons. The residuals based on the ensemble mean GRACE spherical harmonics
display a strong north-south signal in the mid North Atlantic, likely due to north-south stripe
noise affecting spherical harmonic solutions (Blazquez et al., 2018).
To further investigate the North Atlantic sea level misclosure, we computed the North Atlantic
sea level budget after geographically averaging each component over the region, using
GRACE mascons for the manometric component. This is shown in Figure 7    , along with the
sea level budget, averaging the data globally but excluding the North Atlantic Ocean.

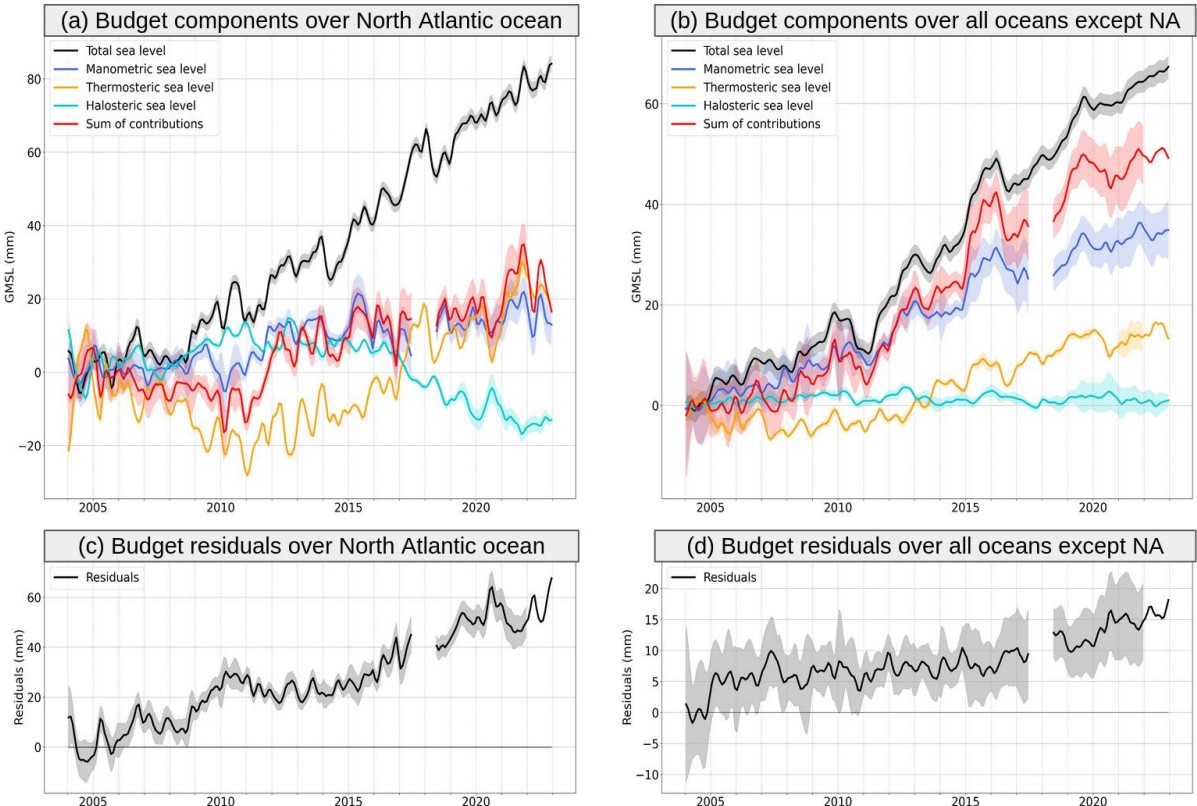


*Figure 7: Regionally averaged sea level budget for January 2004 to December 2022, over the*
*North Atlantic Ocean (a) and over all oceans except the North Atlantic (NA) one (b). On each*
*panel are shown the altimetry-based sea level (black curve), the thermosteric and halosteric*
*components (orange and turquoise curves), the manometric component (GRACE mascons,*
*blue curve) and the sum of all components (red curve). The panels c and d show the*
*corresponding residuals (observed sea level minus sum of components). Shaded areas*
*represent the standard one-sigma uncertainties.*
Figure 7 well confirms the non-closure of the budget over the North Atlantic Ocean, with a
significant positive residual trend, whereas in the remaining oceanic domain, no significant
residual trend is noticed. Figure 7 (panel a) suggests that the North Atlantic residual trend is
related to the observed decrease of the halosteric component as of 2015   . A similar finding
is provided in Mu et al. (2024).


In order to check whether the North Atlantic residual signal better fits a trend over the study
period rather than a low frequency oscillation, we performed an EOF decomposition over
2004-2022 of the gridded residual time series (considering the GRACE SH solution for the
manometric component) (see Figure S3 in the Supplementary Information, showing the first
two EOF modes). Mode 1 is dominated by a strong residual trend in the North Atlantic. Its
spatial map is very similar to the residual map. Mode 2 shows an oscillation of period ~11
years on which are superimposed shorter fluctuations related to ENSO. This EOF
decomposition of the residuals confirms the dominant trend contribution of the North Atlantic
over the study period.

# 634  7. Conclusion

In this study, we have revisited the regional sea level budget over the GRACE and Argo era.
Using different data sets for the manometric component (GRACE and ocean reanalyses), we
found significant non-closure of the budget in the North Atlantic Ocean in all studied cases.
However, the residual patterns are not localized over the same areas in the GRACE and ocean
reanalyses cases. They are stronger in the northeast Atlantic Ocean when considering the
GRACE manometric component (mascon solution) while they are more localized in the
northwestern tropical part with the reanalysis-based manometric component. The sea level
budget averaged over the whole North Atlantic Ocean leads us to suspect the steric
contribution, especially the halosteric component as the main contributor to the budget non-
closure in this region, considering that the global budget without the North Atlantic region is
closed within the error bars. Although we chose the SIO data set to estimate the steric
component, considering that the salinity data had been corrected for the Argo floats
instrumental drift that led to spurious salinity measurements, our study points to remaining
errors affecting the halosteric component, especially as of 2015. Mu et al. (2024) also report
a potential salinity bias in the Argo data set of the North Atlantic. Our results suggest that the
salinity adjustment to the WOCE salinity climatology proposed by the SIO methodology may
not fully correct for the rapid salinity drift experienced by some Argo floats. However, the
different locations of the North Atlantic residual patterns when considering the reanalyses or
GRACE for the manometric component, as well as larger residuals in the northeast Atlantic
Ocean in the GRACE case, suggest that uncertainty in GRACE data also plays a role. In other
oceanic regions, a few areas display      small-scale residual structures (e.g., in the north and
eastern Pacific and northwest Indian Ocean). This may eventually result from differences in
resolution of the gridded data sets used in this study (e.g., satellite altimetry better resolves
small scale features than GRACE or Argo), even though the same  low pass filter was applied
to all data sets.
The problem of the North Atlantic halosteric component highlighted in our regional budget
study needs to be fixed up at the processing level of the Argo-based measurements. Our
findings are helpful to the scientific groups involved in the Argo network as we can identify
regions where the salinity contribution to regional sea level change appears to be spurious.
This may help them in refining their investigations on the quality control checks to be applied
to the Argo profiles. In addition, detailed investigation of the GRACE contribution to the
residuals of this region should also be carried out in parallel. Our study highlights the necessity
of applying consistent data processing and using similar reference systems for satellite
altimetry and gravimetry data. Improved data should indeed be made available to the
community, not only for sea level budget assessments but also for other applications in
oceanography or climate-related research (i.e., for Earth's energy imbalance studies based
on sea level budget approaches).

## 672 **Acknowledgements**

We thank two anonymous reviewers for their helpful comments. We also thank the
Magellium/LEGOS climate change team for providing us with the ensemble mean of the
GRACE-based spherical harmonics solutions and fruitful discussions.
This research was carried out under a programme of, and funded by, the European Space
Agency (ESA) Climate Change Initiative, within the project entitled "Sea level budget closure
CCI+ (SLBC_CCI+)" (contract number 4000140620/23/I-BN).    Part of this work is a
contribution to the GREAT project which is supported by the Centre National d'Etudes
Spatiales (CNES) through the Ocean Surface Topography Science Team (OSTST).    Part of
this work was also performed in the context of the ERC Synergy GRACEFUL project (ERC
Synergy                        grant                        n°                        855677).



## Authors contribution

Conceptualization: MB, AB (Barnoud), AC; Data analysis: MB, AB (Barnoud); Reanalysis computation: CY, AS: Writing of the original draft: AC; Final writing, review and editing: all co-authors.

## Conflict of interest

The authors declare no conflict of interest relevant to this study.

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

# Supplementary Information

**I.** **Trends and trend uncertainties of altimetry-based total sea level, components and budget residuals over 2004-2022**

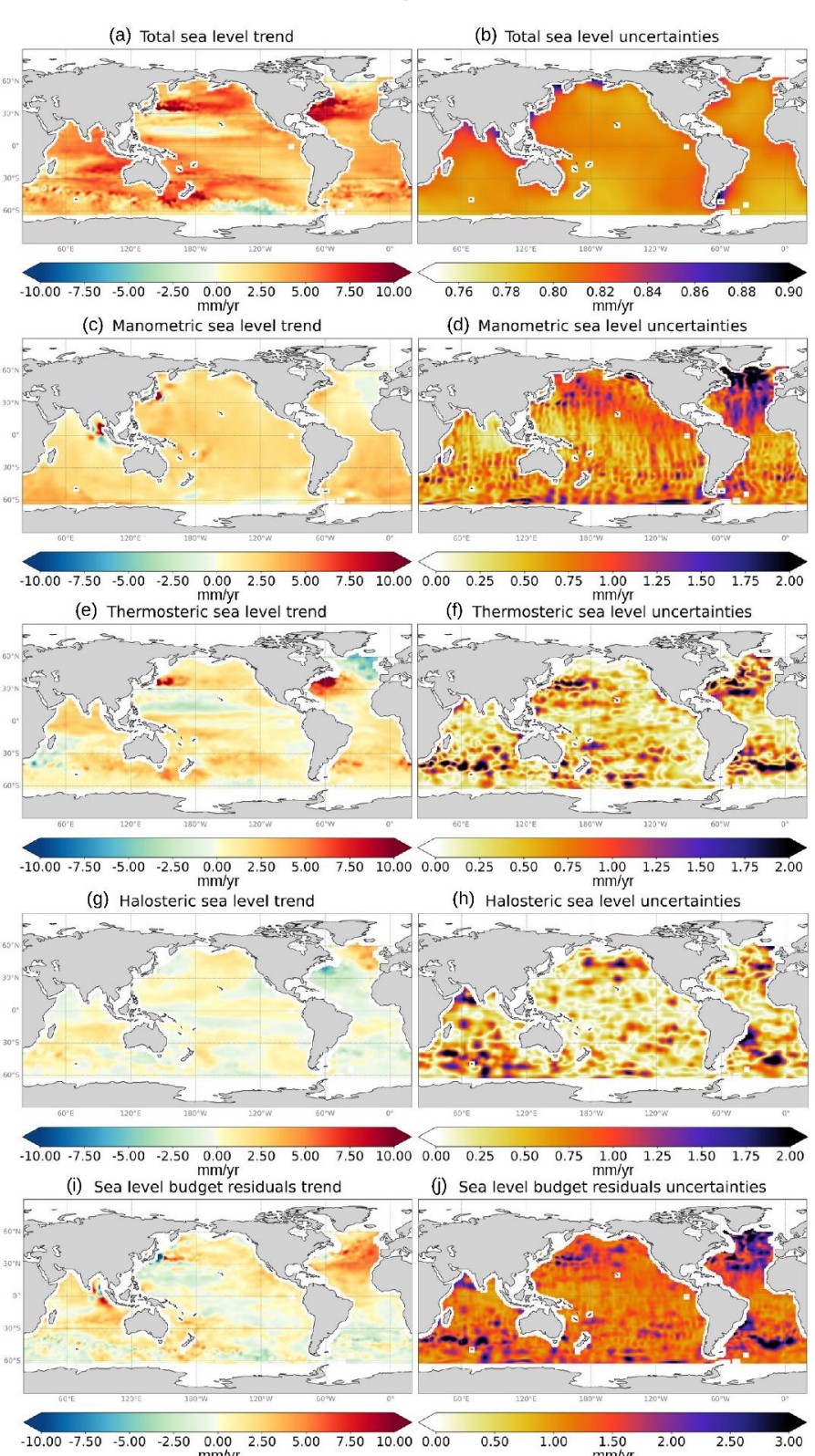

*Figure S1: Trend map (left panels) and associated 1-sigma trend uncertainty map (right panels) over 2004-2022 for each term of the regional sea level budget: (a,b) Total altimetry-based sea level trends and associated trend uncertainties, (c,d) GRACE-based manometric component and associated trend uncertainties, (e,f ) thermosteric component and associated trend uncertainties, (g,h) halosteric component and associated trend uncertainties, and (i,j) budget residuals and associated trend uncertainties.*

## II. Sea level budget residuals of the low degree harmonics of the components

In order to highlight the contribution of the different low degree harmonics in the sea level budget residuals, we decomposed each of the components, restricted to their common oceanic mask, in spherical harmonics and computed the sea level budget for different combinations of these low degree harmonics. The sum of all components summed up to degree 4 (Figure S2, top left panel) well reproduces the sea level budget residual map characteristics with high residuals in the North Atlantic Ocean. Other panels of Figure S2 show the residuals of the sea level budget with components summed up to degree 4, with one harmonic signal corresponding to one (degree l, order m) combination. Table S1 provides the root mean squares (RMS) for each case, computed over all oceans. Figure S2 and Table S1 show that residuals are strongly reduced in the North Atlantic Ocean when removing harmonics l1m0 (geocenter term), l1m1, l2m1 (including the polar motion term), l3m2, l3m3 and l4m2. This means that these harmonics contain spurious signals in some of the sea level budget components. Globally and in the North Atlantic Ocean, harmonics l1m0, l2m1 and l4m3 have the highest impact on the residuals.

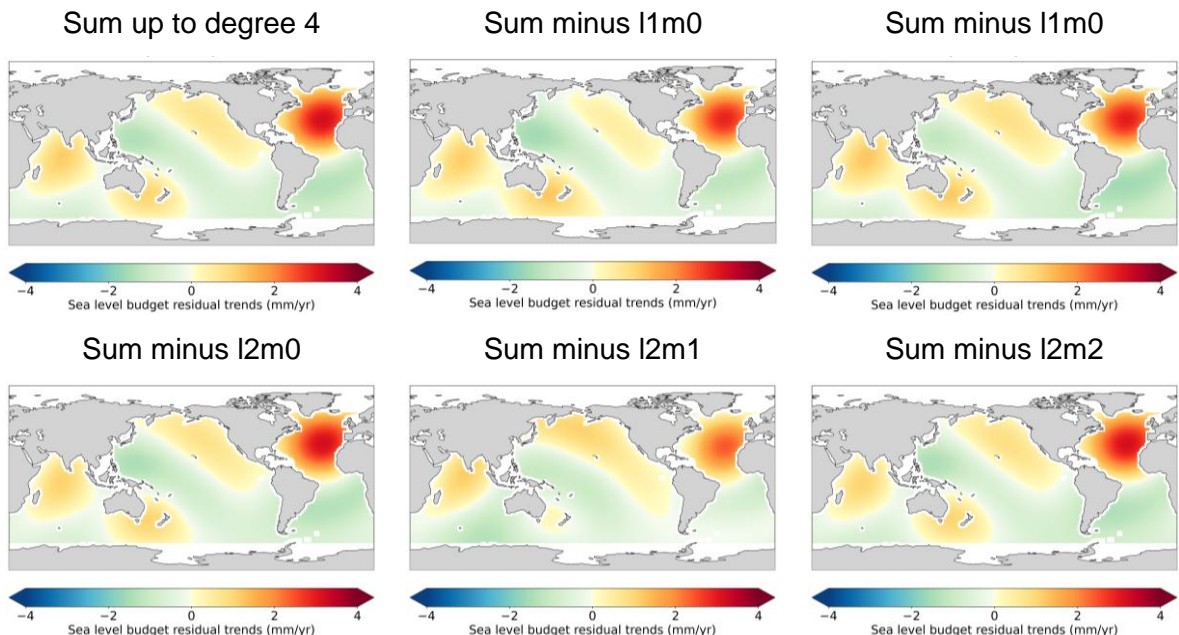

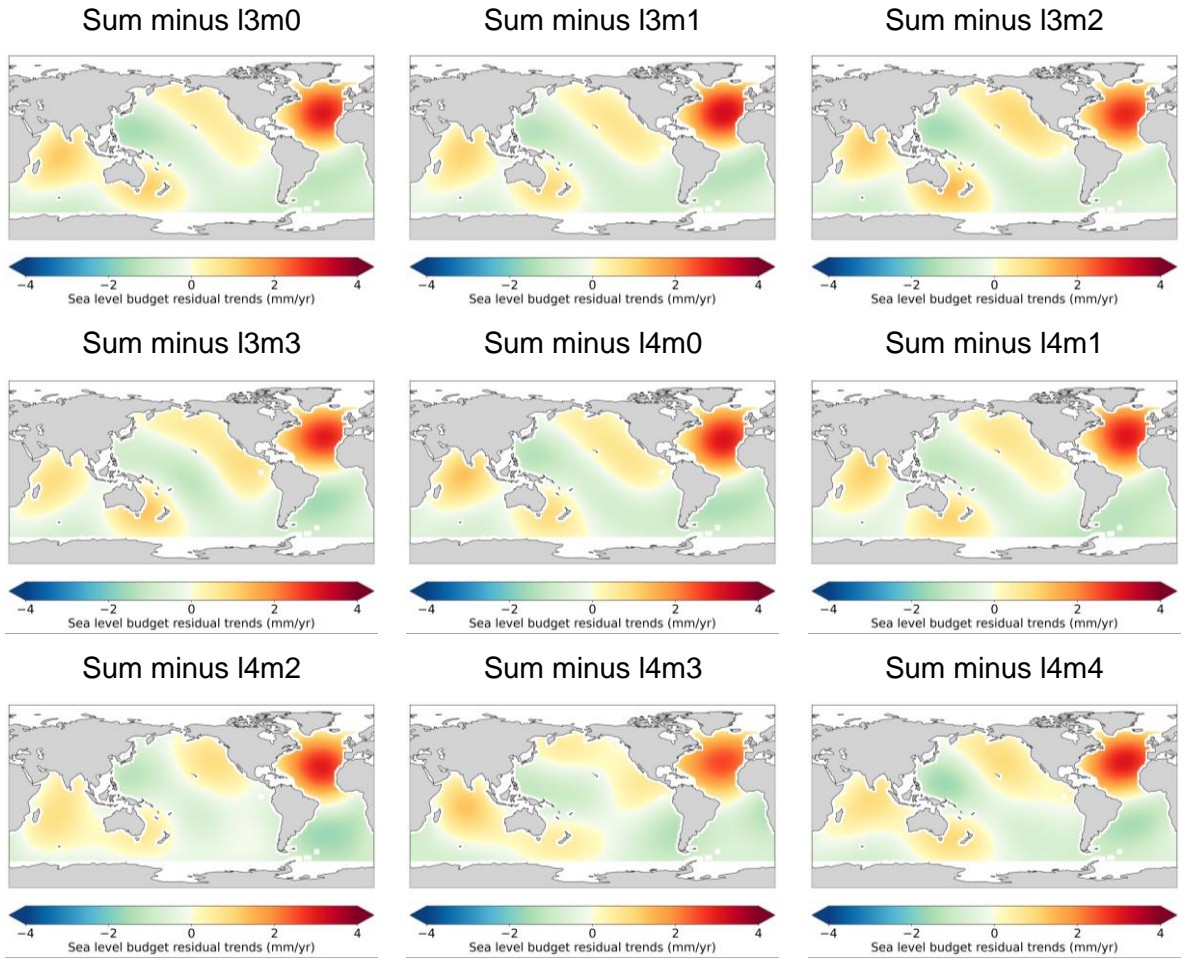

*Figure S2: Sea level budget residual trends computed over January 2005 to June 2022 using*
*the low degrees of each component up to degree 4, and subtracting each order/degree*
*contribution one by one.*
*Table S1: Root mean square (RMS) of the residual trends computed using the low degrees of*
*each component up to degree 4, and subtracting each order/degree contribution one by one.*
*The RMS is computed over the global oceans and over the North Atlantic, South Atlantic,*
*Indian and Pacific Oceans. Bold font indicates RMS values which are lower than 0.1 mm/yr*
*below the value for the sum up to degree 4 without any subtraction (first line).*

| Component removed from the sum of all components up to degree 4 | Residual trends RMS over oceans (mm/yr) | | | | |
|---|---|---|---|---|---|
| | Global | North Atlantic | South Atlantic | Indian | Pacific |
| None | 0.94 | 2.11 | 1.00 | 0.55 | 0.64 |
| l1m0 | 0.87 | **1.86** | **0.87** | 0.54 | 0.65 |
| l1m1 | 0.92 | **1.91** | 1.18 | 0.60 | 0.60 |
| l2m0 | 0.94 | 2.11 | 0.99 | 0.51 | 0.65 |
| l2m1 | **0.74** | **1.50** | **0.48** | 0.69 | 0.56 |

| | | | | | |
|---|---|---|---|---|---|
| l2m2 | 0.93 | 2.08 | 0.96 | 0.51 | 0.64 |
| l3m0 | 0.92 | 2.01 | 0.92 | 0.59 | 0.64 |
| l3m1 | 0.93 | 2.13 | 0.98 | 0.50 | 0.61 |
| l3m2 | 0.93 | **1.93** | **0.76** | 0.58 | 0.76 |
| l3m3 | 0.90 | **1.95** | 1.04 | 0.46 | 0.63 |
| l4m0 | 0.94 | 2.08 | 1.04 | 0.61 | 0.62 |
| l4m1 | 0.91 | 2.03 | 0.97 | 0.52 | 0.61 |
| l4m2 | 0.90 | 2.03 | 1.16 | **0.43** | **0.52** |
| l4m3 | 0.84 | **1.80** | **0.80** | 0.64 | 0.57 |
| l4m4 | 0.93 | 2.07 | 1.03 | 0.47 | 0.65 |

## III. EOF Decomposition of the gridded residual time series (with the GRACE SH-based manometric component)

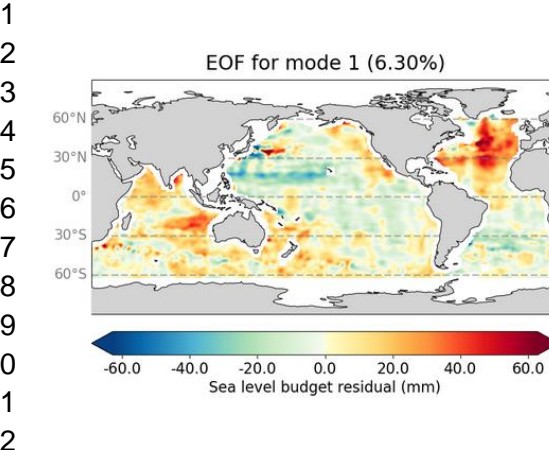
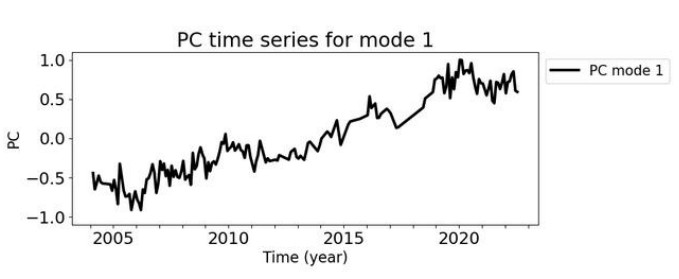

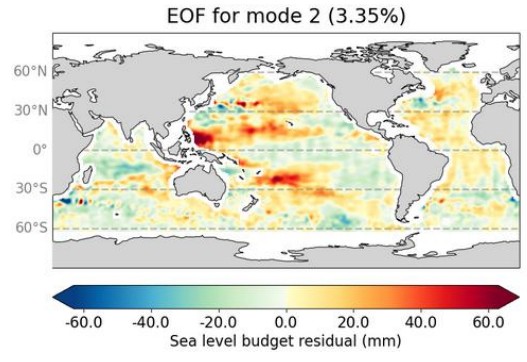
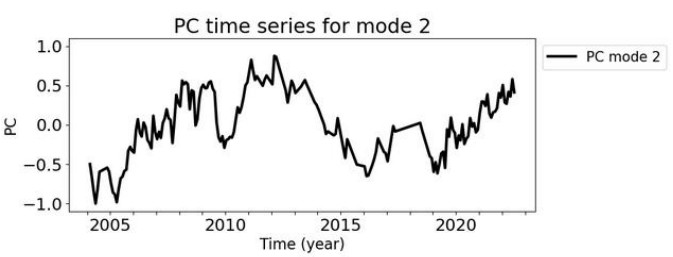

*Figure S3: Modes 1 and 2 of the EOF decomposition over 2004-2022 of the gridded residual*
*time series (with the GRACE SH manometric component). The left panels are the spatial maps*
*while the right panels are the associated principal components (PC).*