# Peer review of "Regional sea level budget"

_EGUsphere, 2024_

## Author Comment (AC1)

**Manuscript « Regional sea level budget over 2004-2022 »**

**Responses to the Reviewer 1 Comments (*in italics*)**

**30 March 2025**

**I. Reviewer 1**

The work under review is a relatively straightforward regional sea level budget analysis based on the Scripps Argo product for the steric component, several gravity-derived and reanalysis-based manometric sea level estimates, and the C3S altimeter dataset for sea level. I have a considerable number of issues that I wish the authors can address in a revision of the current draft. Major and minor comments are listed below, with reference to line numbers where relevant.

*Major comments*

The paper deals with linear trends in many of the figures but there is no discussion of whether the trends are statistically significant (i.e., statistically different from zero at some confidence level). This seems to be a major shortcoming of the presentation and can be simply corrected by stippling or not plotting insignificant values in figures 1—7. The discussion should then be focused on statistically significant non-zero trends.

*Response*

*We agree with Reviewer 1 that this was a major gap in the initial version of the manuscript. We have now added trend uncertainty maps for the sea level and components and focus the discussion on regions where the trend signal in all terms of the budget and residuals is significant (i.e., above the noise).*

In addition, all budget analyses need to carry at least a qualitative discussion of possible uncertainties in all the terms, in this case sea level, steric and manometric components derived from the different products. The authors use an ensemble of manometric estimates but they fail to provide a spread or some quantitative measure of uncertainty for their results.

*Response*

*See above*

And some features in their results are treated lightly or ignored. For example, there is a clear issue with the major Tohuku and Sumatra earthquakes affecting the trends in the mascon solutions, but that is never mentioned.

*Response*

*We now have briefly discussed the signal related to the Sumatra and Tohuku earthquakes in the GRACE-based manometric map and residuals (based on GRACE)*

Last but not least, I find the discussion of many figures loosely presented and at places not really jiving with what I could reasonably infer from the figures themselves. Here are several examples of this issue:

325-326/ I would disagree. For example, halosteric term is larger than thermosteric term in parts of the South Indian Ocean, northeast North Pacific, and manometric term is larger than thermosteric in parts of the Southern Ocean.

*Response*

*The text has been modified*

333-335/ I think the claimed similarity between manometric and thermosteric terms is hardly justified, and actually I can't think of any reason why we should expect them to be similar.

*Response*

*We did not say that. However to be clearer, we rephrased this sentence.*

336-339/ The text mentions strong residuals, but contrary to what it is claimed, in the eastern Atlantic there are relatively strong negative thermosteric trends, not fully compensated by positive halosteric trends, which seem to give rise to the large residuals (sea level trends are fairly weak);

*Response*

*Text has been modified for clarity*

419-420/ Again, I am having a hard time seeing this claim in the figure. Actually, I see opposite signs in spatial trends of the ensemble mean reanalyses and GRACE in the North Atlantic as well.

*Response*

*We did not say that trends in GRACE and reanalyses-based manometric components have the same sign. They have indeed opposite sign in many regions, including in the North Atlantic (that part of the sentence 'except in the North Atlantic…' was an error. It is now corrected).*

If qualitative claims are going to be made, the authors should make sure they are clearly evident in the figures. In addition, quantitative analyses such as determining pattern correlation coefficients should be attempted when deriving inference about similarity between spatial trend patterns.

*Response*

*Quantitative results are provided in Table 2 that gathers RMS residuals for each reanalysis and GRACE mascon cases and for each oceanic region.*

**Minor comments**

51-52/ Note that "redistribution" can also lead to changes in steric component (you can redistribute density, not just mass).

*Response*

*Text modified*

66-60/ This sentence (or perhaps actually the full paragraph) almost equates global mean sea level budget with the issue of global mean sea level rise, but of course budget analyses can and should apply to all time scales, not just the linear trend, for both global mean and regional cases. A recent example of regional budgets for the seasonal cycle is found in https://doi.org/10.1029/2024EA003978.

*Response*

*This study focuses on trends. Looking at the annual budget is beyond the scope of the present study*

101/ Delete "already in the ocean".

*Response*

*Done*

141-143/ Sentence could be improved for clarity.

*Response*

*Done*

157-158/ Statement is not strictly true, as changes in global mean pressure can come from changes in the mean water vapor content of the atmosphere and those can affect the barystatic term. Likely a small effect, however!

*Response*

*The water vapor contribution essentially affects the global mean barystatic component, which is removed in the present analysis.*

168/ "GRDs" sounds weird! "GRD effects" or similar would read better, here and elsewhere in the text.

*Response*

*Corrected*

171-172/ I think this statement pertains to "absolute" sea level. Please clarify the text. Similar issues may apply to other parts of the text.

*Response*

*The word 'absolute' is added when needed*

175/ Delete "sea level"; also "northeast coast of North America".

*Response*

*Done*

205/ "which leads to"

*Response*

*Corrected*

179/ "based on observations"

*Response*

*Corrected*

213-214/ I think you are referring to the drift in the global mean, but statement needs to be clarified.

*Response*

*It seems that it is was we wrote*

217-218/ We need some estimate of these deep steric changes at the local scales of interest to this paper.

*Response*

*We agree with the reviewer's comment. We added the following sentence to emphasize our statement : "Based on deep Argo profiles, Lele and Purkey (2024) estimated the deep ocean steric sea level rise (temperature and salinity contribution) being 0.13 ± 0.16 mm yr$^{-1}$ in the south Pacific Ocean over 2014-2023".*

246/ "two filtering levels"? Please clarify this text.

*Response*

*We removed this part of the sentence because this concerns to detailed technical parts of the data processing.*

254-255/ The GRACE and GRACE-FO records have many gaps. Please clarify in the text how those are handled in the analyses.

*Response*

*We added a sentence explaining how gaps are taken into account*

263-264/ Unclear sentence.

*We have rewritten the sentence for clarity.*

283-285/ Can the authors discuss, at least qualitatively, what sort of errors this may imply in the derived manometric trends?

*Considering a linear extrapolation for the fingerprints at the end of the record may have negligible impact on the reanalysses-based manometric component considering that this contribution is very smal*

287-293/ I am confused by the treatment of atmospheric loading corrections for GRACE. Of course, GRACE does not "see" atmospheric loading effects, if one has an inverse barometer behavior.  Effects of atmospheric loading would only be apparent in the global mean bottom pressure, but those are apparently removed in the present analyses. I think this needs to be clarified, to make sure corrections are appropriately applied.

*Response*

*Text has been clarified*

329-333/ You start by calling out large residuals in the North Atlantic but those are in the eastern part of the basin. You go on to discuss the western part and south of Greenland, where residuals actually seem relatively small. This is somewhat confusing.

*Response*

*Text has been modified*

340-345/ It would be useful to include an extra panel with residuals calculated on the basis of steric trends only. Do they look better? Actually including such panel in figure 2 might justify that figure better. Otherwise, figure 2 is not needed, as those same two panels can be readily examined in figure 1.

*Response*

*The steric trend map is now added*

374/ "everywhere" is an overstatement given results in figure 3c,d.

*Response*

*Corrected*

375/ Very unclear what inferences are being made in this text, given the previous discussion of results in figure 3. Please rewrite for clarity.

*Response*

*Text has been modified*

399/ Delete either " 's" or "the" before Camargo.

*Response*

*Corrected*

434-435/ "CIGAR also does not assimilate"

*Response*

*Corrected*

447/ "If we exclude FOAM"

*Response*

*Corrected*

459/ "assimilation in Figure 4 (e.g.,"

*Response*

*Corrected*

458-463/ You reference figure 6 but the text seems to be comparing CIGAR and C-GLORS results in figure 5?

*Response*

*Corrected*

461/ The largest differences are actually west of the Drake Passage.

*Response*

*A sentence has been added*

463-465/ I don't follow the corollary.

*Response*

*Text has been corrected*

475/ Actually all reanalyses, not just CIGAR show positive residuals.

*Response*

*Text has been corrected*

494/ "stripe"

*Response*

*Corrected*

495-497/ Very unclear what this sentence means. Please rewrite.

*Response*

*Text has been corrected*

Figure 9/ This figure is hardly justified. It does not bring anything new to what is already discernible from Figure 8a. I don't think the global EOF adds any relevant information to the discussion.

*Response*

*Figure 9 has been deleted*

511-513/ I think the residual for "all but North Atlantic" case is also significant?

*Response*

*The sum of components agrees reasonably well with the sea level curve. Even if the residual curve is not perfectly flat, it does not show the strong decrease as in the North Atlantic case.*

514/ The halosteric decrease is evident after 2016, not 2013-2014?

*Response*

*Yes this is true. Corrected*

---

## Author Comment (AC2)

**Manuscript « Regional sea level budget over 2004-2022 »**

**Responses to the Reviewer 2 Comments (*in italics*)**

**30 March 2025**

**II. Reviewer 2**

Review for: "Regional sea level budget over 2004-2022" by Marie Bouih, Anne Barnoud, Chunxue Yang, Andrea Storto, Alejandro Blazquez, William Llovel, Robin Fraudeau and Anny Cazenave  (https://doi.org/10.5194/egusphere-2024-3945).

The authors investigate the regional closure of 20-year sea level trend budget based on altimetry (total), ARGO (steric), GRACE/Reanalysis (manometric) sea level datasets and GRD fingerprints. The paper focuses on the differences between various manometric datasets. It examines the influence of the GRACE processing on the regional sea level budget and emphasises the need to apply the geocenter correction. The largest regional trend discrepancies occur in the North Atlantic, and the authors suggest that a spurious drift in the salinity measurements may be responsible.

The paper deals with a relevant topic and is well written. The introduction provides a good overview of the topic and cites the relevant literature. Equations could help to make it easier to find out which components were taken into account for which data sets to calculate the residual trends. The figures are clear and informative, however, some of them seem to be in the paper twice. If possible, the datasets and the corresponding versions should be cited unambiguously (e.g. doi for altimetry).

Unfortunately, none of the manometric components studies provides a sufficient closure of the regional trend budget.  Possible causes for the differences between the datasets should be discussed in more detail.  A discussion of the uncertainties of the trend estimates is missing for all data sets. With regard to the North Atlantic, the explanation for not closing the trend budget should be better justified. The figures suggest rather decadal variability between the gyres than consistent long-term drifts in the entire area.

*Response*

*We have added maps of trend uncertainties for the sea level data, components and residuals, and now focus on the regions where the residual signal is above the noise.*

*We also performed an EOF analysis of the gridded residual time series (the corresponding figure has been added in the SI). It is reproduced below (using GRACE SH solution for the manometric component):*

[Figure]

*Figure S3: Modes 1 and 2 of the EOF decomposition over 2004-2022 of the gridded residual time series (with the GRACE SH manometric component). The left panels are the spatial maps while the right panels are the associated principal components (PC).*

*Mode 1 is dominated by a strong residual trend in the North Atlantic. Its spatial map is very similar to the residual map. Mode 2 shows a low frequency oscillation of period around 11 years on which are superimposed shorter fluctuations related to ENSO.*

**Specific comments:**

Lines 78-92:

The usage of the expressions regional, basin-scale, sub-basin scale and local scale is confusing and could even be inconsistent

*Response*

*These terms were introduced to distinguish between the different spatial scales*

Lines 141-143: Shouldn't the barystatic component distribute according to the GRD fingerprints?

*Response*

*The barystatic term is spatially quasi uniform. It was removed for the regional budget assessment.*

Line 180: Could you provide numbers for the small regional GIA & contemporary GRD trends?

*Response*

*Added*

Line 190-200: Please specify the data version (doi?). Some of the given information is abundant since it is not used (Topex side A drift, total sea level uncertainties)

*Response*

*Added*

Lines 217: Could you provide numbers for the regional deep steric contributions?

*Response*

*Added*

Lines 265-: Could you specify the differences between the manometric data from ocean reanalysis? What is the uncertainty and why did you choose these models?

*Response*

*Added*

Line 271: Are all reanalyses based on ARGO data? Is it known how the salinity drift is handled in the individual reanalyses?

*Response*

*This is indeed a key issue. We added the following text:*

*"One may wonder whether the salinity drift observed in some Argo floats as of 2015 is impacting the CIGAR reanalysis since, unlike altimetry data, T/S data are assimilated during the reanalysis integration, thus non-linearly interacting with dynamical processes. The treatment of the salinity drift simply consisted in rejecting data that Argo had flagged for rejection in the delayed mode. But this may not fully guarantee that all bad salinity data have been discarded. However, to compute the reanalysis-*

*based manometric component, the local steric contribution is removed. Thus any*

*effect of the Argo salinity drift should be minimized."*

Lines 304-305: Figure 1 suggests that the spatial filtering of altimetry and mascon datasets is not consistent.

*Response*

*The same filtering is indeed applied. However due to the lower resolution of GRACE data compared to altimetry, the GRACE-based manometric map looks smoother.*

Lines 306-308: Are there systematic differences between the trends for these two periods?

*Response*

*No there are no significant trend differences. But adding 3 years is interesting since it shows that the trends do not change, hence do not reflect short term fluctuations.*

Figure 2: Is there a difference to figure 1b and 1e?

*Response*

*Figure 2 has been deleted since it was redundant with Figure 1*

Line 340-345, Couldn't there be problems with the degree 2 terms of the GRACE-processing as well?

*Response*

*There is a consensus within the GRACE community to use such values.*

Lines 370-390: I would suspect that the ARGO-data, as well as the ocean reanalyis, are referenced to the centre of figure.  Even though the altimeter orbits were calculated relative to the centre of mass they may have been transferred  to the centre of figure somewhere on their way to the level 4  Copernicus sea level grids.

*Response*

*Normally, all data are expressed in the center of figure reference frame  but for the altimetry data, the way the geocenter correction is performed may is still an issue (Alexandre Couhert, personal communication)*

Figure 4: Trends differ quite a lot. What is the uncertainty of individual  trend estimates?

*Response*

*There is no information on individual reanalysis uncertainty. Only dispersion around the ensemble mean can be estimated.*

Line 458: You decide to focus on the CIGAR model, but the results for CIGAR are not included in Table 2.

*Response*

*This has been added*

Lines 511-515: The halosteric component seems to be on the decline after 2015/2016 and to be anticorrelated with the thermosteric component. You might want to consider the budget for the subtropical and the subpolar gyre separately.

*Response*

*We leave this for a future work that will focus on the North Atlantic*

Why should the effects of a spurious drift in salinity measurements only be effective in the North Atlantic?

*Response*

*Because it is in this region that most instrumental drifts have been detected*

Figure 9: If you suspect that the PC1/EOF1 of the halosteric signal is dominated by the North Atlantic signal, why do you perform a global EOF-analysis?

*Response*

*The figure has been deleted*

---

## Author Comment (AC3)

**Manuscript « Regional sea level budget over 2004-2022 »**

**Responses to C. Ludwigsen Comment (*in italics*)**

**30 March 2025**

**Comment posted by Carsten Ludwigsen**

This paper addresses a critical aspect of the regional sea level budget. Beyond the reviewers' comments**, I believe it's essential to discuss how reanalysis models assimilate Argo data and how potential issues in the Argo dataset might propagate into the reanalysis outputs.** Given this, I find the suggestion that GRACE may not accurately observe manometric sea level is currently insufficiently supported. The authors should provide a more detailed elaboration on these points to strengthen their analysis.

*Response*

*This is indeed a key issue. We added the following text:*

*"One may wonder whether the salinity drift observed in some Argo floats as of 2015 is impacting the CIGAR reanalysis since, unlike altimetry data, T/S data are assimilated during the reanalysis integration, thus non-linearly interacting with dynamical processes. The treatment of the salinity drift simply consisted in rejecting data that Argo had flagged for rejection in the delayed mode. But this may not fully guarantee that all bad salinity data have been discarded. However, to compute the reanalysis-based manometric component, the local steric contribution is removed. Thus any effect of the Argo salinity drift should be minimized."*

---

## Referee Report (RR1)

Review for: "Regional sea level budget over 2004-2022" by Marie Bouih, Anne Barnoud, Chunxue Yang, Andrea Storto, Alejandro Blazquez, William Llovel, Robin Fraudeau and Anny Cazenave (https://doi.org/10.5194/egusphere-2024-3945).

The paper and the figures have been significantly improved. However, additional adjustments could further improve the overall quality. The authors may wish to consider the following comments:

- Even though the estimation of trend uncertainties is not in the focus of this paper, the uncertainties are still crucial for the interpretation of the residual trends. Do you include formal fit errors in your trend estimats? For Figures 5 and 6, one could assume that the uncertainty of the CIGAR reanalysis is similar to the uncertainty of the whole ensemble and add this information to the corresponding plots.
- Subsection 4 is a bit confusing and may need a revision. For comparisons between GRACE and altimetry, the GRACE data is commonly transferred to the centre of figure (e.g. https://doi.org/10.1029/2004GL020461). You question whether the altimeter data are actually given in the centre of figure and conclude that they appear to be. You also mention possible inconsistencies in the modelling of polar motion in GRACE and altimetry data. Could you add a few sentences to explain the possible inconsistencies in more detail?
- The largest regional trend residuals occur in the North Atlantic, and the authors suggest that a spurious drift in the salinity measurements after 2015 may be responsible. This conclusion is not very well supported. Figure 7c shows that there is a residual drift throughout the whole time series rather than a drift starting in 2015. This should be discussed further.

**Specific comments:**

Lines 143-145: Since the barystatic component is related to mass changes on land, the water should not be distributed evenly but according to the fingerprint-pattern. However, these are very small numbers.

Lines 383-386 & lines 395-397: Now you mention signal of the large earthquakes I the GRACE mascons twice.

Lines 429-430: Check sentence: The results confirm the referential of altimetry ?

Lines 512-513/545: In fact, the residual trend patterns are similar, even though the manometric components do not agree. The dominant feature seems to be differences between the total and steric components that cannot be compensated by the rather small manometric component.

Lines 561/562: wouldn't the circulation estimate be shifted due to the spurious salinity values?

Supplement:

- Figure S1: could be improved by using same colorbar for trends as in figures 1 - 6, and the same color bar for all uncertainty-subplots
- Figure S2, Table S1: It is difficult to relate the low degree spherical harmonics to the actual physical drivers. Do different versions of the low degree terms give different results?

---

## Author Response (AR2)

**RESPONSES TO THE REVIEWERS**

**I.       Responses to Reviewer 1 (in italics)**

 Review for: "Regional sea level budget over 2004-2022" by Marie Bouih, Anne Barnoud, Chunxue Yang, Andrea Storto, Alejandro Blazquez, William Llovel, Robin Fraudeau and Anny Cazenave (https://doi.org/10.5194/egusphere-2024-3945).

The paper and the figures have been significantly improved. However, additional adjustments could further improve the overall quality. The authors may wish to consider the following comments:

- Even though the estimation of trend uncertainties is not in the focus of this paper, the uncertainties are still crucial for the interpretation of the residual trends. Do you include formal fit errors in your trend estimats? For Figures 5 and 6, one could assume that the uncertainty of the CIGAR reanalysis is similar to the uncertainty of the whole ensemble and add this information to the corresponding plots.

*Response*

*- As explained in the text (lines 344-346 of the previous revised version, the uncertainties of the residuals are based on the square root of the sum of squares of errors of each component. Uncertainties of altimetry is based on Prandi et al. (2021) while for the steric and manometric components, they are based on the dispersion of different products around their mean? This is also explained in the text (page 12 of the previous revised version).*

*- For CIGAR, we prefer not use the uncertainty of the whole ensemble because this may not be appropriate for a single reanalysis.*

- Subsection 4 is a bit confusing and may need a revision. For comparisons between GRACE and altimetry, the GRACE data is commonly transferred to the centre of figure (e.g. https://doi.org/10.1029/2004GL020461). You question whether the altimeter data are actually given in the centre of figure and conclude that they appear to be. You also mention possible inconsistencies in the modelling of polar motion in GRACE and altimetry data. Could you add a few sentences to explain the possible inconsistencies in more detail?

*Response*

*We agree with Reviewer 1 that the text was not clear enough. We replace the previous text (lines 429 to 434 of the previous revised version) by the following:*

*"Let's remind that GRACE data are classically corrected for the geocenter motion when compared with altimetry data, in order to moveing GRACE observations from the centre of mass to the centre of figure of the reference system, in which the altimetry-based sea level is supposed to be also expressed after correcting the satellite orbits for the geocenter motion (Alexandre Couhert, personal communication)."*

- The largest regional trend residuals occur in the North Atlantic, and the authors suggest that a spurious drift in the salinity measurements after 2015 may be responsible. This conclusion is not very well supported. Figure 7c shows that there is a residual drift throughout the whole time series rather than a drift starting in 2015. This should be discussed further.

*Response*

*We also agree with Reviewer 1 that in Figure 7c, the residual time series presents a positive trend over almost the whole period (same for Figure 7d). But there is a clear shift as of 2015 that may be linked to the negative trend in the halosteric component (Figure 7a). We modified the text accordingly.*

**Specific comments:**

Lines 143-145: Since the barystatic component is related to mass changes on land, the water should not be distributed evenly but according to the fingerprint-pattern. However, these are very small numbers.

*Response*

*Text has been clarified*

Lines 383-386 & lines 395-397: Now you mention signal of the large earthquakes I the GRACE mascons twice.

*Response*

*Text has been clarified*

Lines 429-430: Check sentence: The results confirm the referential of altimetry ?

*Response*

*Text has been clarified (see above response to the general comment)*

Lines 512-513/545: In fact, the residual trend patterns are similar, even though the manometric components do not agree. The dominant feature seems to be differences between the total and steric components that cannot be compensated by the rather small manometric component.

*Response*

*We agree with the Reviewer's comment*

Lines 561/562: wouldn't the circulation estimate be shifted due to the spurious salinity values?

*Response*

*A sentence has been added*

Supplement:

- Figure S1: could be improved by using same colorbar for trends as in figures 1 - 6, and the same color bar for all uncertainty-subplots

*Response*

*We did that on purpose. Using the same color bar for all maps will lead to unreadable figures (some will be too flat, and for others, colors will be saturated)*

- Figure S2, Table S1: It is difficult to relate the low degree spherical harmonics to the actual physical drivers. Do different versions of the low degree terms give different results?

*Response*

*We are unsure about what Reviewer 1 means by "different versions of low degree terms" (different combinations? different mask?). Some physical drivers (geocenter motion, polar motion, GIA) particularly impact specific degrees. However, looking at the sea level budget residuals, i.e. at all sea level components, we are restricted to the oceanic domain and not the whole Earth, making this link less straightforward. The*

*use of low degrees here is meant to help identifying potential sources of spurious signals in the sea level budget residuals, hence we used combinations showing the impact of each degree on the residuals.*

**II.    Responses to Reviewer 2 (in italics)**

The authors have satisfactorily addressed the issues raised in my original review and I am happy to recommend publication at this point. There are a few minor typos that I have tried to list below. I also follow up on two previously raised points, for the consideration of the authors when creating a final version of the manuscript.

67/ My original comment on lines 66-70 did not request treatment of seasonal cycle budgets in the paper, as implied by the authors' response, but basically wanted to highlight the issue of equating "sea level budgets" with "sea level trend budgets". This issue is apparent all over the manuscript, starting with the title, which in my view would be clearer if it read "Regional sea level trend budgets over 2004-2022". The addition of "Focusing on trends" at the beginning of the revised paragraph on line 67 does address part of the issue, but I would suggest that the authors explicitly state somewhere that reference to "sea level budget" in the manuscript means specifically "trend budgets", or else use the latter expression where appropriate (especially in section titles).

*Response*
*We added 'trend' everywhere in the manuscript*

81/ "ocean basin-scales" or "the ocean basin-scale"

*Response*
*Corrected*

92/ "at the ocean basin-scale or smaller.

*Response*
*Corrected*

202/ "estimated to be"

*Response*
*Corrected*

278/ "steric effect from the"

*Response*
*Corrected*

335/ State explicitly what "dispersion" is and how it is calculated.

*Response*
*We modified the text for clarity*

415/ "in order to move"

*Response*
*Corrected*

545-546/ "similar. Figure 5 shows manometric

*Response*
*Corrected*

617-619/ As in my original comment, if we look at figure 7d, the residual is significantly different from zero. Thus, if I am reading figure 7d correctly, I think this sentence is somewhat misleading and would be better rephrased. If the authors wanted to go a little further, what I find interesting is that while the residual in figure 7d is mostly flat up until around 2015, it trends upward substantially after 2015, in fact

similar to what happens in the North Atlantic (figure 7c). Thus, the analyses suggest similar relevant issues with the regional trend budgets outside the North Atlantic, although only noticeable when multi-basin averages are used as in figure 7d.

*Response*
*We modified the text as follows:*

*"Figure 7 well confirms the non-closure of the budget over the North Atlantic Ocean, with a significant positive residual trend, whereas in the remaining oceanic domain, no significant residual trend is noticed. Figure 7 (panel a) suggests that the North Atlantic residual trend is related to the observed decrease of the halosteric component as of 2015 of the North Atlantic."*